# Hedgehog regulation of epithelial cell state and morphogenesis in the larynx

Janani Ramachandran[1], Weiqiang Zhou[2], Anna E Bardenhagen[1], Talia Nasr[3,4], Ellen R Yates[1], Aaron M Zorn[3,4], Hongkai Ji[2], Steven A Vokes[1]*

[1]Department of Molecular Biosciences, The University of Texas at Austin, Austin, United States; [2]Department of Biostatistics, Johns Hopkins Bloomberg School of Public Health, Baltimore, United States; [3]Center for Stem Cell and Organoid Medicine (CuSTOM), Division of Developmental Biology, and Perinatal Institute, Cincinnati Children's Hospital Medical Center, Cincinnati, United States; [4]Department of Pediatrics, University of Cincinnati College of Medicine, Cincinnati, United States

**Abstract** The larynx enables speech while regulating swallowing and respiration. Larynx function hinges on the laryngeal epithelium which originates as part of the anterior foregut and undergoes extensive remodeling to separate from the esophagus and form vocal folds that interface with the adjacent trachea. Here we find that sonic hedgehog (SHH) is essential for epithelial integrity in the mouse larynx as well as the anterior foregut. During larynx-esophageal separation, low *Shh* expression marks specific domains of actively remodeling epithelium that undergo an epithelial-to-mesenchymal transition (EMT) characterized by the induction of N-Cadherin and movement of cells out of the epithelial layer. Consistent with a role for SHH signaling in regulating this process, *Shh* mutants undergo an abnormal EMT throughout the anterior foregut and larynx, marked by a cadherin switch, movement out of the epithelial layer and cell death. Unexpectedly, *Shh* mutant epithelial cells are replaced by a new population of FOXA2-negative cells that likely derive from adjacent pouch tissues and form a rudimentary epithelium. These findings have important implications for interpreting the etiology of HH-dependent birth defects within the foregut. We propose that SHH signaling has a default role in maintaining epithelial identity throughout the anterior foregut and that regionalized reductions in SHH trigger epithelial remodeling.

*For correspondence:
svokes@austin.utexas.edu

Competing interest: The authors declare that no competing interests exist.

## Editor's evaluation

The authors present cellular and genetic data, combining mutant analysis and RNA-sequencing, that together support a functional role for Shh in repressing the epithelial-to-mesenchymal transition (EMT) in the developing larynx during larynx-esophageal separation. In the absence of Shh, cells undergo EMT and are replaced with a novel epithelial cell population of unknown origin. These results make a significant contribution to the field by illuminating how the larynx develops.

## Introduction

The larynx produces all of the sounds for vocal communication and regulates swallowing and access to the esophagus and trachea that lie directly beneath it. Congenital laryngeal malformations such as tracheo-laryngeal clefts and bifid epiglottis arise from defects in early laryngeal morphogenesis and impair tracheo-esophageal function (feeding and breathing), as well as vocalization in infants, often requiring surgical intervention and significantly impacting patients' quality of life (*Biesecker, 1993*; *Cohen et al., 2006*; *Johnston et al., 2014*; *Leboulanger and Garabédian, 2011*). Complicating the etiology of these disorders, the pathways that drive the early stages of larynx morphogenesis,

specifically vocal fold closure and larynx-esophageal separation, remain largely unknown. Several recent findings suggest that HH signaling may be important for early larynx development. Early loss of HH signaling results in the reduction of SOX2-expressing cells from the larynx epithelium and a failure in vocal fold closure (*Lungova et al., 2015*). HH signaling also drives the separation of the trachea and esophagus, which are directly caudal to the larynx (*Billmyre et al., 2015*; *Han et al., 2020*; *Ioannides et al., 2010*; *Kim et al., 2019*; *Kuwahara et al., 2020*; *Nasr et al., 2019*; *Que et al., 2007*). Mutations in the HH pathway transcriptional effector GLI3 cause dramatically altered larynx morphology and vocalization defects (*Tabler et al., 2017*). Similarly in humans, laryngeal clefts and bifid epiglottis are phenotypes of Pallister Hall syndrome which arises from truncating mutations in GLI3 (*Biesecker, 1993*; *Böse et al., 2002*; *Ondrey et al., 2000*). Together these defects suggest that HH signaling may be required for several stages of larynx morphogenesis beyond vocal fold closure.

The larynx is derived from the early foregut epithelium which is regionally differentiated into multiple organs, including the pharynx, parathyroid, thymus, trachea, esophagus, and larynx in the anterior half. Induction of these organs from the nascent gut tube, as well as subsequent morphogenesis are driven by specialized types of epithelial remodeling such as budding, branching, septation, and epithelial-to-mesenchymal transitions (EMTs) (*Bort et al., 2006*; *Bort et al., 2004*; *Hebrok, 2000*; *Hogan and Kolodziej, 2002*; *Qi and Beasley, 2000*). These are regulated by localized signaling interactions, including HH, between the foregut and the surrounding mesenchyme (*Han et al., 2020*; *Han et al., 2017*; *Jacobs et al., 2012*; *Kraus and Grapin-Botton, 2012*; *Nerurkar et al., 2017*; *Rankin et al., 2016*; *Zorn and Wells, 2009*). In the anterior foregut, organogenesis is uniquely affected by the influx of migratory neural crest-derived cell populations that combine with populations of meso-dermally derived mesenchymal cells to form region-specific pharyngeal structures (*Bain et al., 2016*; *Bhatt et al., 2013*; *Brito et al., 2006*; *Kuo and Erickson, 2010*; *Tabler et al., 2017*; *Trainor and Tam, 1995*).

The larynx arises from an unknown cellular origin at the base of the pharynx adjacent to the fourth pharyngeal pouches (*Figure 1A*), and bridges the anterior-most portions of the foregut to the more posterior trachea and esophagus (*Essien and Maderious, 1981*; *Henick, 1993*; *Lungova et al., 2015*; *Poopalasundaram et al., 2019*). The early stages of larynx development are characterized by three major epithelial remodeling events, beginning with the stratification and zippering of the lateral walls of the foregut along the midline to close the vocal folds and form the epithelial lamina. Within the next 24 hr, the epithelial lamina, which joins the dorsal esophagus and the ventral trachea, puckers to form the infraglottic duct, and separates from the esophagus (*Henick, 1993*; *Lungova et al., 2015*). The newly separated lamina then fully recanalizes to form a laryngeal lumen that is continuous with the trachea, around which specialized cartilage elements and musculature are specified (*Henick, 1993*; *Lungova et al., 2018*; *Lungova et al., 2015*). Epithelial morphogenesis is genetically dependent upon WNT, HIPPO, and HH signaling although the underlying mechanisms remain poorly understood (*Lungova et al., 2018*; *Lungova et al., 2015*; *Mohad et al., 2021*; *Tabler et al., 2017*).

We asked if and how HH signaling might regulate epithelial remodeling during larynx development. We defined distinct domains of epithelium that downregulate *Shh* and undergo EMT-based remodeling during larynx-esophageal separation and esophageal constriction. We uncovered a similar process in *Shh*$^{-/-}$ embryos, in which epithelial cells lose expression of canonical foregut genes and undergo an EMT marked by cadherin switching and ultimately cell death. Despite massive cell death, the anterior foregut retains a rudimentary epithelium that now contains an ectopic population of cells. These findings provide a cell-based mechanism for understanding previously defined HH-dependent vocal fold closure and laryngeal cleft defects (*Lungova et al., 2015*). As similar changes are seen beyond the larynx, we propose a model in which regionalized reductions in HH drive dynamic epithelial remodeling throughout the anterior foregut.

## Results

### Larynx epithelial cells downregulate *Shh* expression and undergo EMT-based remodeling during larynx-esophageal separation, and esophageal constriction

To determine if *Shh* might regulate epithelial remodeling in the larynx, we examined its expression at E11.75, when the vocal folds are remodeled to separate the larynx from the esophagus. There

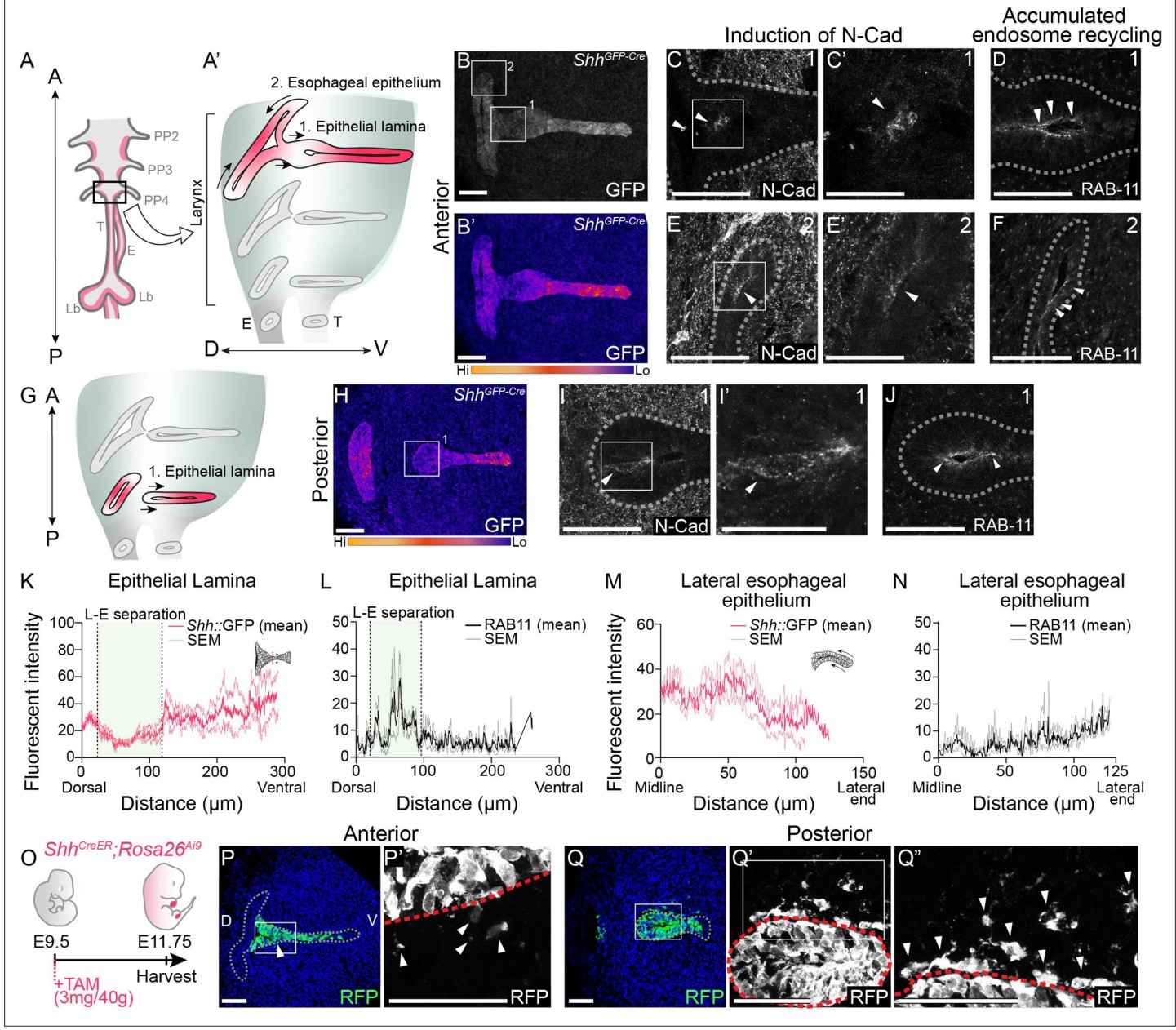

**Figure 1.** Actively remodeling epithelial cells have low *Shh* expression and undergo epithelial-to-mesenchymal transition (EMT) during larynx-esophageal separation and esophageal constriction. Schematic of the anterior foregut (**A**) highlighting the anterior (**A'**) and posterior (**G**) larynx at E11.75. (A, B, H) GFP marking *Shh* expression in the anterior and posterior larynx (*n* = 5 for each). There is reduced GFP expression at the epithelial lamina which fuses and then separates the larynx and esophagus (region 1; **B, H**), and the constricting esophageal opening (region 2; **B**) of the larynx. (**C, E–E', I–I'**). Expression of N-Cadherin at anterior or posterior regions at E11.75 (*n* = 3). Arrowheads mark N-Cadherin expression in the epithelium. RAB-11 was visualized at regions 1 (**D, J**) and 2 (**F**) in three larynxes. Arrowheads mark apical RAB-11 expression in the epithelium. Relative GFP expression along the epithelium at regions 1 (**K**) and 2 (**M**) was measured and averaged across three replicates by line scans of fluorescent intensity. Standard error of mean was calculated across all three replicates and plotted in light pink. Relative RAB-11 expression was measured by line scans of fluorescent intensity along the epithelium at regions 1 (**L**) and 2 (**N**) and averaged across three replicates. Standard error of mean was calculated across all three replicates and plotted in gray. (**O–Q**). Shh-descendant cells were visualized in three E11.75 larynxes using *Shh^CreER*;*Rosa26^Ai9* lineage labeling. *Shh^CreER/+*;*Rosa26^Ai9/+* embryos were induced with Tamoxifen at E9.5 and analyzed for RFP (green) expression (**P–Q**) at E11.75 along the anterior–posterior axis of the larynx. Arrowheads mark *Shh* descendants in the mesenchyme. Panels (**B, B', H, P, and Q**) are single slice images. All other panels are z-projections. A – anterior; P – posterior; D – dorsal; V – ventral (panels **A–J** and **P–Q** are in the same orientation); PP 2/3/4 – pharyngeal pouches #2–4; Lar – larynx; T – trachea; E – esophagus; Lb – lung buds. (**C', E', I'**). Scale bars denote 25 µm. All other scale bars denote 50 µm.

The online version of this article includes the following figure supplement(s) for figure 1:

*Figure 1 continued on next page*

Figure 1 continued

**Figure supplement 1.** *Shh* expression is reduced, and E-Cadherin is re-localized in the epithelial lamina and the constricting esophagus during larynx-esophageal separation.

**Figure supplement 2.** *Shh*-descendant cells express N-Cadherin and undergo epithelial-to-mesenchymal transition (EMT) during larynx-esophageal separation.

was a wide variation in *Shh* expression within the larynx, with markedly reduced domains of *Shh*^GFP expression in the epithelial lamina at the future site of larynx-esophageal separation in addition to the lateral edges of the esophagus that are in the process of constricting (*Figure 1A, B, G, H, K, M*). The regional reduction in GFP reporter expression is corroborated by a reduction in *Shh* gene expression in both regions (*Figure 1—figure supplement 1A–D*), as well as the absence of *Shh*-descendant cells from these regions (*Figure 1—figure supplement 2A, B*).

Because *Shh* was reduced in both regions of the larynx undergoing dynamic epithelial remodeling, and previous studies observed *Shh*-descendant cells in the mesenchyme directly adjacent to larynx-esophageal separation (*Lungova et al., 2018*), we asked whether low *Shh* expression in the larynx epithelium was accompanied by cadherin switching and EMT. Consistent with this possibility, membranous N-Cadherin was expressed within the epithelial layer both in the epithelial lamina adjacent to the infraglottic duct at the site of larynx-esophageal separation, as well as along the lateral edges of the constricting esophagus (*Figure 1C, E*). N-Cadherin-expressing cells were also present in more posterior sections of the separated larynx, overlapping with the region of reduced GFP expression (*Figure 1I*). While there was no overall reduction in E-Cadherin protein (*Figure 1—figure supplement 1E–I*), there was an increase in punctate E-Cadherin expression in both regions (*Figure 1—figure supplement 1H′, I′*). The re-localization of E-Cadherin and the concomitant initiation of N-Cadherin at these regions provide evidence for a cadherin switch both at the epithelial lamina and along the lateral edges of the esophagus. This is further supported by the apical accumulation RAB-11, a marker of endosome recycling that is required for the transport of E-Cadherin as well as N-Cadherin to the apical cell surface in multiple contexts (*Desclozeaux et al., 2008*; *Kawauchi et al., 2010*; *Nasr et al., 2019*; *Welz et al., 2014*; *Woichansky et al., 2016*; *Figure 1D, F, J, L, N*).

To determine whether N-Cadherin-expressing cells undergo EMT within these domains we examined *Shh*-descendant cells at E11.75 using a *Shh*^CreER;*Rosa26*^Ai9 reporter line (*Figure 1O*). Consistent with prior reports using a related *Shh*^Cre strategy (*Lungova et al., 2018*), there were a small number of RFP-labeled cells within the mesenchyme along the anterior–posterior axis of the separating larynx and esophagus (*Figure 1P, Q*). While there was a significant increase in the domain of N-Cadherin expression within the remodeling epithelium at later stages of larynx-esophageal separation (*Figure 1—figure supplement 2D–F*), there was no significant increase in the number of mesenchymal RFP-labeled cells (*Figure 1—figure supplement 2B, C*). This suggests that some *Shh*-descendant cells undergo EMT-based extrusion during larynx remodeling but they do not remain in the mesenchyme. Overall, these findings indicate that *Shh* expression is dynamically regulated in the remodeling larynx, with low levels of *Shh* expression coinciding with cadherin switching. The change in cadherin status is likely the underlying cause for the epithelial cells to leave the epithelium.

## Early larynx epithelial cells undergo EMT in the absence of HH signaling

The results so far indicated that regional downregulation of *Shh* was correlated with cadherin switching and EMT. To investigate this further, we generated RNA-seq datasets for control and *Shh*^−/− larynx tissues and identified differentially regulated genes and enriched pathways (*Supplementary file 1*). Consistent with our model, EMT was the most significantly enriched pathway among HH-dependent genes (*Figure 2A*), supporting a role for HH signaling in regulating this process. Differentially expressed genes consisted of members of all three progressive EMT stages (*Figure 2B*; *Lamouille et al., 2014*). These included downregulation of the pro-epithelial adhesion genes *Dsp*, and *Dcn*, which mark the first stage (*Bax et al., 2011*; *Huang et al., 2012*; *Kowalczyk and Nanes, 2012*; *Wang et al., 2015*; *Yilmaz and Christofori, 2009*). There was also an upregulation of the pro-migratory genes *Cdh2*, *Vimentin*, and *Fn1*, indicative of the next phase of EMT (*Wheelock et al., 2008*; *Yilmaz and Christofori, 2009*). Finally, there was a downregulation of *Lama1*, which encodes for Laminin, suggestive of a breakdown in the basement membrane which is one indicator of the third stage of

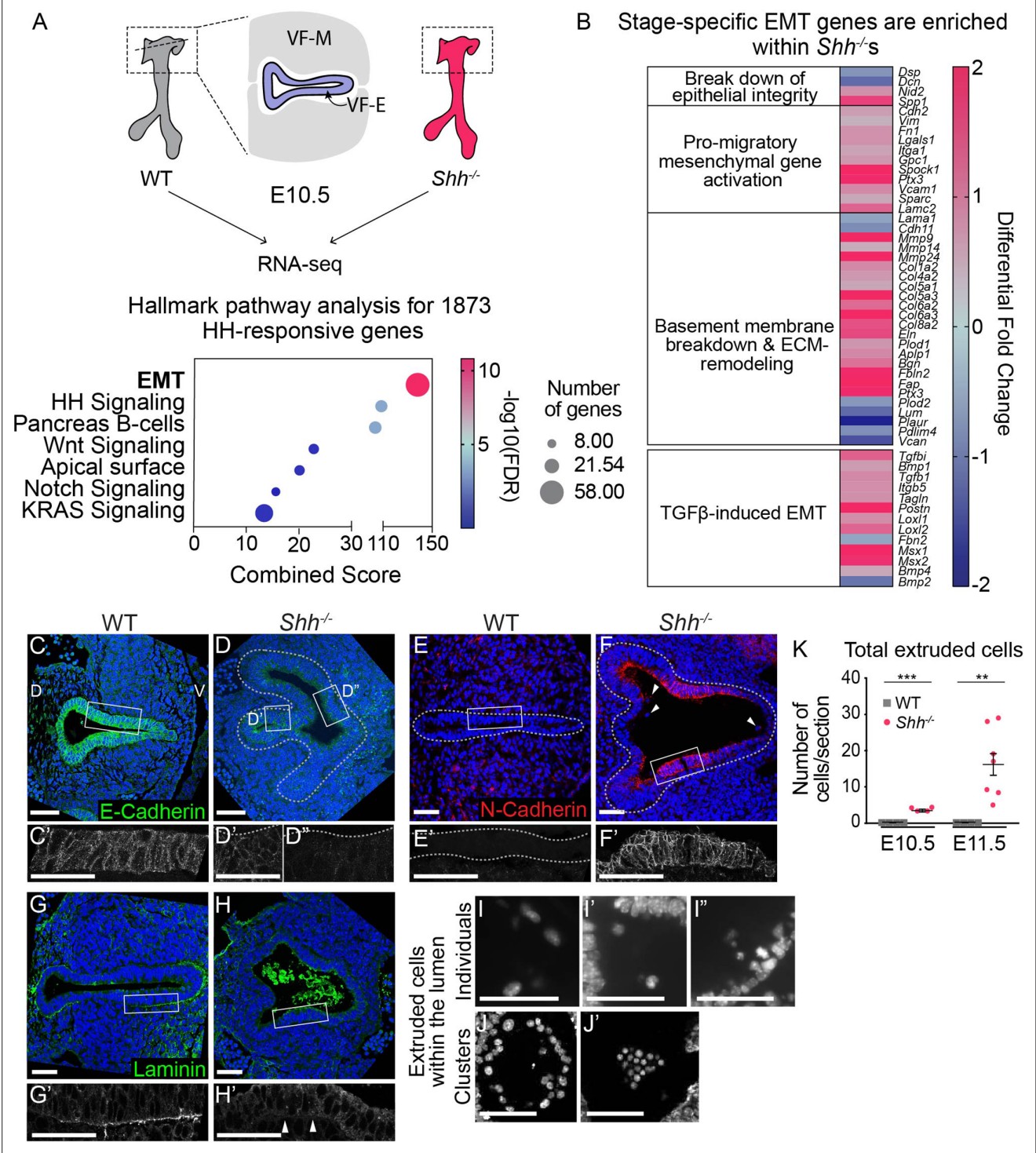

**Figure 2.** Larynx epithelial cells undergo ectopic epithelial-to-mesenchymal transition (EMT)-like cell extrusion in the absence of HH signaling. (**A**). RNA-seq of wild-type (WT) and *Shh*⁻ᐟ⁻larynx tissue at E10.5 identified 1873 HH-dependent genes (false discovery rate (FDR) < 0.05). EMT-related genes were highly enriched among HH targets by Hallmark pathway analysis. (**B**) Differentially expressed EMT genes cluster into stage-specific groups. (**C, D**) E-Cadherin (green) expression in the epithelium across three controls and three *Shh*⁻ᐟ⁻s at E10.5. (**E, F**) N-Cadherin (red) expression within the epithelium of three controls and three *Shh*⁻ᐟ⁻s. Arrowheads mark cells in the lumen. (**G, H**) Laminin (green) expression marking the basement membrane in three controls and three *Shh*⁻ᐟ⁻s. Arrowheads indicate loss of Laminin from the basement membrane in *Shh*⁻ᐟ⁻s. (**I–K**) 4′,6-Diamidino-2-phenylindole (DAPI) staining marking cells within the lumen of the larynx in *Shh*⁻ᐟ⁻s at E10.5 and E11.5. (**K**) Total number of luminal cells/section were quantified in four controls and four *Shh*⁻ᐟ⁻s at E10.5 and in four controls and seven *Shh*⁻ᐟ⁻s at E11.5. Average numbers of luminal cells/section were analyzed for statistical

*Figure 2 continued on next page*

*Figure 2 continued*

significance using the Student's *t*-test. Error bars show the standard error of the mean.**p < 0.005, ***p < 0.0005. VF-M – vocal fold mesenchyme; VF-E – vocal fold epithelium; D – dorsal; V – ventral (panels A–H are in the same orientation). Panels **C', D', D", E', F', G', H'** are z-projections. All other panels are single slices. All scale bars denote 50 μm.

The online version of this article includes the following source data and figure supplement(s) for figure 2:

**Source data 1.** List of all differentially expressed genes detected by RNA-seq in wild-type (*n* = 2 sets of 3-pooled larynx samples) compared to *Shh*⁻/⁻(*n* = 2 sets of 3-pooled larynx samples) pooled larynx samples at E10.5 (32–35 s) .

**Figure supplement 1.** A loss of p63 expression in *Shh*⁻/⁻ laryngeal epithelia.

EMT (*Aumailley and Smyth, 1998*; *Lamouille et al., 2014*; *Nakaya et al., 2008*; *Thiery and Chopin, 1999*). In addition, several TGFβ family members were upregulated, suggesting a plausible mechanism for inducing EMT (*Barrallo-Gimeno and Nieto, 2005*; *Katsuno et al., 2013*; *Mercado-Pimentel and Runyan, 2007*; *Nawshad et al., 2004*; *Schnaper et al., 2003*; *Thiery et al., 2009*).

Consistent with the RNA-seq data and evocative of the observations in remodeling epithelia (*Figure 1*), N-Cadherin (*Cdh2*) was highly upregulated within the mutant epithelium along with substantial reduction of pro-epithelial E-Cadherin (*Cdh1*) and p63 (*Figure 2C–F*; *Figure 2—figure supplement 1A–C*). This suggested that loss of epithelial stratification was accompanied by a shift in the adhesive properties of the larynx epithelium toward a mesenchymal profile. This change was further accompanied by a loss of Laminin from the basement membrane along the epithelium (*Figure 2B, G, H*) indicating that HH signaling is required to maintain the integrity of this structure. Additional basement membrane component genes such as *Col4a2* and *Nid2* were upregulated by RNA-seq, suggesting that loss of Laminin may result in a compensatory increase in other basement membrane components in order to maintain membrane integrity (*Jones et al., 2016*; *Salmivirta et al., 2002*; *Figure 2B*). Interestingly, single cells appeared to be extruding from the epithelial layer into the lumen (*Figure 2I–I"*) and clusters of cells were present within the lumen of the epithelium (*Figure 2J, J'*). These clusters were first seen at E10.5 and increased dramatically by E11.5 (*Figure 2K*). Overall, these observations are consistent with laryngeal epithelial cells undergoing EMT in the absence of HH signaling.

## HH signaling prevents a cadherin switch within the epithelium during early stages of foregut development

To determine the onset of this phenotype, we examined earlier stages of foregut development and found a reduction in the levels of E-Cadherin in the *Shh*⁻/⁻ epithelium of the presumptive larynx as early as E9.5 (*Figure 3A, B*). Although N-Cadherin was not initially expressed at high levels, expression of membranous N-Cadherin was first observed in a small number of cells within the epithelial layer at E9.5 and at E9.75 (*Figure 3C, D, N*). Over the next few hours of development E-Cadherin re-localized within cells, moving from high expression along the lateral cell boundaries to accumulating in puncta along the apical surface (*Figure 3E–G*), but was maintained in the epithelium as late as E10.0 (*Figure 3J, K*; *Aiello et al., 2018*; *Woichansky et al., 2016*). At this stage, more than half of the cells within the epithelium expressed robust levels of membranous N-Cadherin (*Figure 3L–N*) suggesting that N-Cadherin is induced in cells simultaneously expressing E-Cadherin at this stage. This is consistent with recent studies that have described coexpression of E-Cadherin and N-Cadherin in cells undergoing EMT (*Aiello et al., 2018*; *Ray and Niswander, 2016*).

The change in cadherin status suggested a transition to a mesenchymal fate in the absence of HH signaling. Alternatively, these cells might be replaced by a different population of N-Cadherin-expressing cells. To distinguish between these possibilities we examined Cadherin expression in larynx epithelial cells using the *Shh*^CreER;*Rosa26*^Ai9 reporter line to label *Shh*-expressing larynx epithelial cells in control (*Shh*^CreER/+;*Rosa26*^Ai9/+) and mutant (*Shh*^CreER/−;*Rosa26*^Ai9/+) embryos. Tamoxifen induction during early stages of foregut development (E8.5) exclusively labeled epithelial cells in the vocal folds at E9.75 in control and mutant embryos (*Figure 3O–Q*). While E-Cadherin and N-Cadherin had mutually exclusive boundaries restricted to the epithelium and mesenchyme, respectively, in controls, they appeared to be coexpressed within a small number of TdT-expressing epithelial cells in *Shh*^CreER/−;*Rosa26*^Ai9/+s at E9.75 (*Figure 3P–Q*; *Figure 3—figure supplement 1A, B, D*). Coexpression was also observed at E10.0, both as distinct apical puncta and laterally along cell–cell boundaries

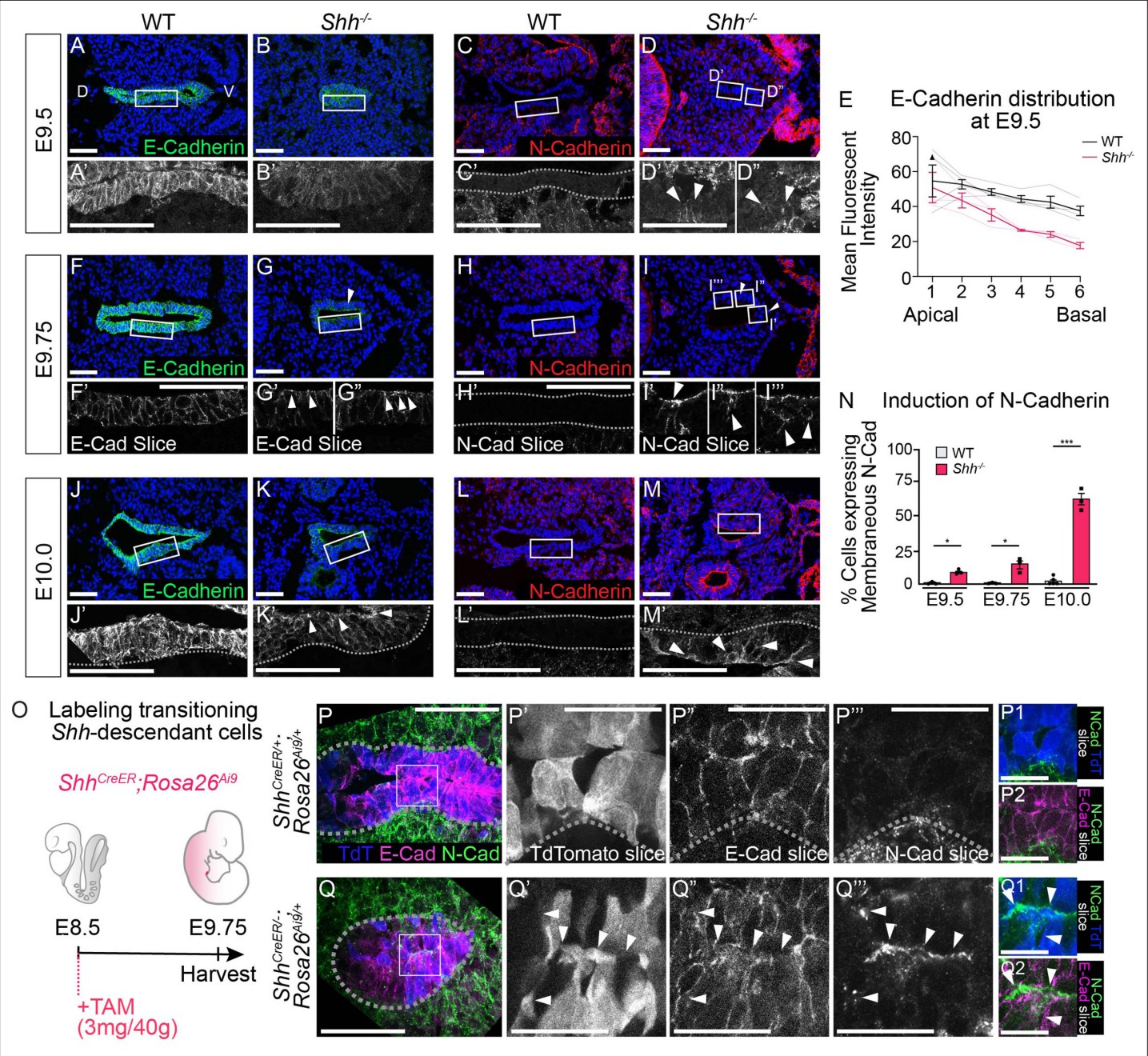

**Figure 3.** HH signaling is required to prevent a cadherin switch within the epithelium during early stages of foregut development. E-Cadherin expression (**A, B**; green) and N-Cadherin expression (**C, D**; red) in four control and three *Shh⁻/⁻* larynxes at E9.5 (24–26 somites). Arrowheads mark N-Cadherin expression in the epithelium. (**E**) E-Cadherin distribution along the apical–basal axis of the epithelium in four controls and three *Shh⁻/⁻*s at E9.5. Error bars show standard error of the mean. E-Cadherin expression (**F, G**; green) and N-Cadherin expression (**H, I**; red) in three control and three *Shh⁻/⁻* larynxes at E9.75 (27–29 somites). Arrowheads mark apical E-Cadherin puncta (**G–G″**) and N-Cadherin expression in the epithelium (**I–I‴**). E-Cadherin expression (**J–K**; green) and N-Cadherin expression (**L–M**; red) was examined in three control and three *Shh⁻/⁻* larynxes at E10.0 (30–31 somites). Arrowheads mark apical E-Cadherin (**K′**) and membranous N-Cadherin expression in the epithelium (**M′**). (**N**) The percentage of N-Cadherin-expressing cells within the epithelium at E9.5, E9.75, and E10.0 was averaged across three to five controls and three *Shh⁻/⁻* larynxes at each stage and analyzed for significance using the Student's *t*-test (*p < 0.05, ***p < 0.0005). Error bars indicate the standard error of the mean. (**O**) *Shh*-fate mapping in control (*Shh^CreER/+^;Rosa26^Ai9/+^*) and mutant (*Shh^CreER/−^;Rosa26^Ai9/+^*) embryos (four replicates each). Td-Tomato (TdT) labeling was induced with Tamoxifen at E8.5 and visualized at E9.75. (**P–Q**) Sections were analyzed for E-Cadherin (E-Cad; magenta) and N-Cadherin (N-Cad; green) expression as well as Td-Tomato (TdT; blue) expression. Arrowheads mark regions of E-Cadherin and N-Cadherin expression along the membrane of a Td-Tomato-positive cell. D – dorsal; V – ventral (all panels are in the same orientation). Panels **A–D, F–I″, J–M, P′–P2, Q′–Q2** are single slice images. Panels **A′, B′, C′, D′-D″, J′, K′, L′, M′, P, Q** are z-projections. **P′–P2, Q′–Q2**. Scale bars denote 25 µm. All other scale bars denote 50 µm.

*Figure 3 continued on next page*

*Figure 3 continued*

The online version of this article includes the following figure supplement(s) for figure 3:

**Figure supplement 1.** Larynx epithelial cells coexpress E-Cadherin and N-Cadherin during early stages of foregut development in the absence of HH signaling.

**Figure supplement 2.** HH is required to maintain *Cdh1* expression in the early anterior foregut.

**Figure supplement 3.** Vimentin and SNAIL in *Shh*$^{-/-}$ cells that leave the epithelium.

(*Figure 3—figure supplement 1C*), indicating that laryngeal epithelial cells undergo a cadherin switch in the absence of HH signaling. The switch in cadherin expression within the vocal folds also occurred on the transcriptional level with *Cdh1* expression in the mutant epithelium at E9.25 replaced by high levels of *Cdh2* expression by E10.5 (*Figure 3—figure supplement 2A–D*).

In some systems, the expression of SNAIL and Vimentin is necessary and sufficient to induce EMT (*Barrallo-Gimeno and Nieto, 2005*; *Casas et al., 2011*; *Huang et al., 2012*; *Jägle et al., 2017*; *Liu et al., 2015*; *Mendez et al., 2010*; *Scheibner et al., 2021*; *Vuoriluoto et al., 2011*). As suggested by the transcriptional increase in pro-migratory factors at E10.5, there was an increase in low-level Vimentin expression along the apical surface of TdT-expressing epithelial cells in *Shh*$^{CreER/-}$;*Rosa26*$^{Ai9/+}$s at E9.75 compared to controls (*Figure 3—figure supplement 3A–B, B1–B1", E*). Vimentin expression also marked a small number of TdT-expressing cells within the mesenchyme in *Shh*$^{CreER/-}$;*Rosa26*$^{Ai9/+}$s at this stage (*Figure 3—figure supplement 3B2–B2"*; 4/12 mesenchymal TdT$^+$ cells). SNAIL expression was not detected within the mutant epithelium (*Figure 3—figure supplement 3C–D, D1–D1'''*, *E*) but approximately half of the TdT-expressing epithelial cells in the mesenchyme were also positive for SNAIL expression (*Figure 3—figure supplement 3D2'–D2'''*; 5/9 mesenchymal TdT$^+$ cells) suggesting that TdT$^+$ cells that migrate into the mesenchyme are capable of transiently expressing mesenchymal cell markers (see Discussion).

FOXA2 activates the expression of *Cdh1* and suppresses EMT programs in the endoderm (*Bow et al., 2020*; *Scheibner et al., 2021*; *Zhang et al., 2015*). This suggested that EMT initiation within *Shh*$^{-/-}$s might be caused by a loss of FOXA2 expression. Consistent with this scenario, FOXA2 was expressed in nearly every cell at the region of the future larynx in both control and *Shh*$^{-/-}$s during early stages of foregut development (E9.25; 21–23 somites; *Figure 4A*, *Figure 4—figure supplement 1A–B*). However, by E9.75 (27–29 somites), FOXA2 expression was not detectable in ~30–40% of the mutant cells with reduced expression in many of the remaining FOXA2$^+$ cells (*Figure 4A–C*, *Figure 4—figure supplement 1C–F*). FOXA2 was further reduced by E10.5 and completely absent by E11.5 while expression remained robust in controls (*Figure 4A*, *Figure 4—figure supplement 1G–J*). This suggested that HH might prevent EMT by positively regulating FOXA2 in either a cell non-autonomous or autonomous fashion. In keeping with the latter possibility, there was *Ptch1* and low-level *Gli1* expression within the larynx epithelium (*Figure 4—figure supplement 2A–C*), indicating that HH signaling could potentially regulate FOXA2 through autocrine signaling. As epithelial remodeling in wild-type embryos occurred in low-*Shh*-expressing cells (*Figure 1*, *Figure 1—figure supplement 1*), we asked if these regions also had downregulated FOXA2. In contrast to the early foregut (*Figure 4A*, *Figure 4—figure supplement 1*), FOXA2 was not downregulated at the site of larynx-esophageal separation or along the lateral edges of the constricting esophagus (*Figure 1*, *Figure 4—figure supplement 3A–F3, G, H, I*). We conclude that FOXA2 regulation by HH signaling is specific to early stages of foregut and larynx development.

## Transitioning epithelial cells extrude from the epithelial layer and undergo apoptosis in the absence of HH signaling

We next asked what happened to foregut epithelial cells undergoing EMT once they left the epithelium. These cells could be in the process of undergoing apoptosis, as often happens with extruded cells (*Fadul and Rosenblatt, 2018*; *Kim et al., 2015*; *Kuipers et al., 2014*; *Ohsawa et al., 2018*). Alternatively, these cells might persist in the mesenchyme and contribute to adjacent developing tissues. To address this, we again used genetic fate mapping to examine the fate of larynx epithelial cells, in control (*Shh*$^{CreER/+}$;*Rosa26*$^{Ai9/+}$) and mutant *Shh*$^{CreER/-}$;*Rosa26*$^{Ai9/+}$ embryos (*Figure 4D*, *Figure 4—figure supplement 4*). Td-Tomato labeling was largely restricted to the epithelial layer in control embryos (*Figure 4E–G, K*, *Figure 4—figure supplement 4A–B*). Consistent with earlier experiments

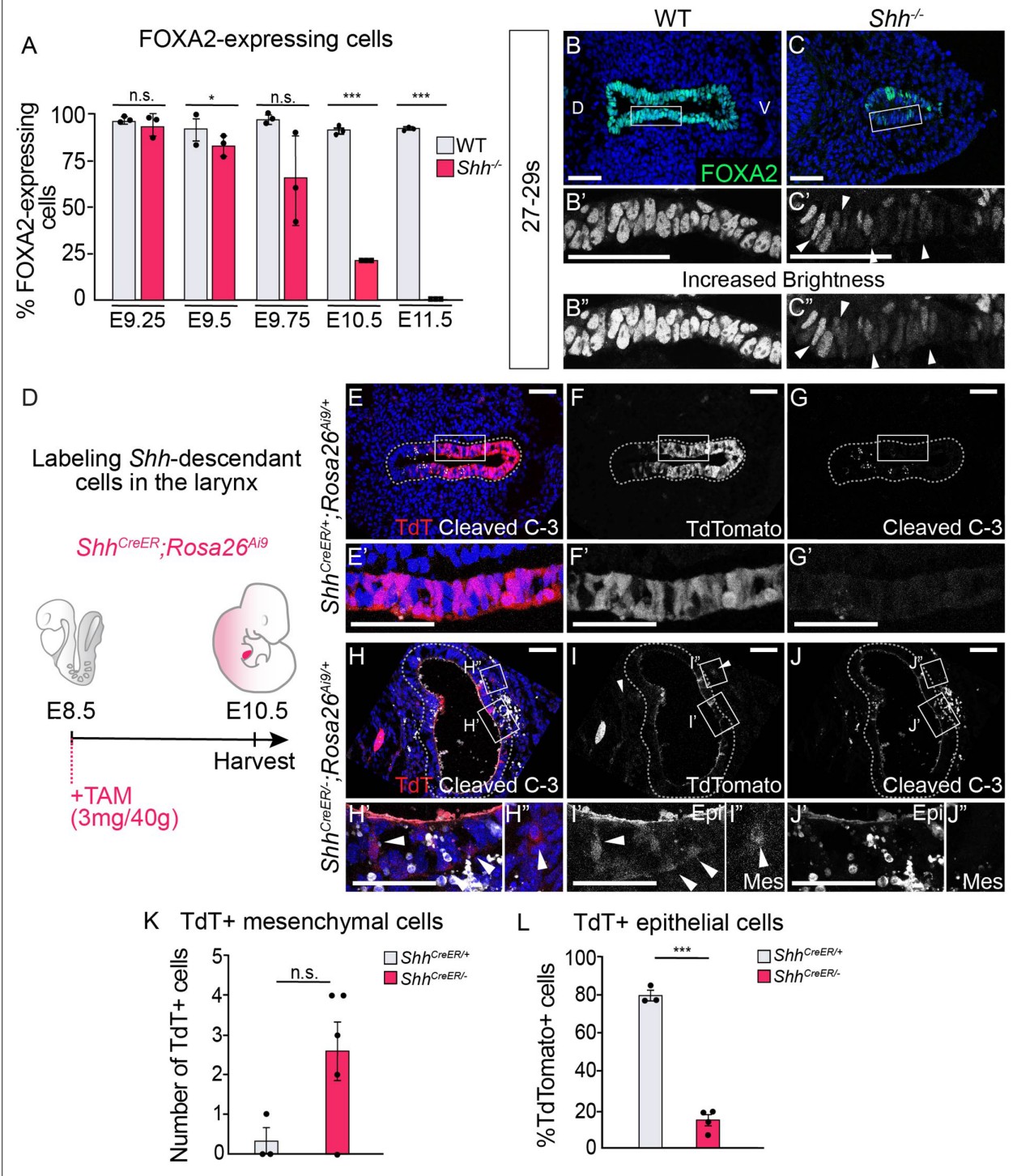

**Figure 4.** Epithelial cells lose FOXA2 and leave the epithelium in the absence of HH signaling. (**A**) FOXA2 expression in control and *Shh⁻/⁻* larynxes at E9.25 (21–23 s), E9.5 (24–26 s), E9.75 (27–29 s), E10.5 (31–35 s), and E11.5 (42–44 s). The percentage of epithelial cells expressing FOXA2 in three control and three *Shh⁻/⁻* larynxes was averaged at each developmental stage and analyzed for significance using a Student's *t*-test. Error bars show the standard error of the mean. (**B–C**) FOXA2 (green) expression is reduced in *Shh⁻/⁻*s by E9.75 (27–29 s) compared to controls. Arrowheads indicate FOXA2-low cells (**B', B''**). Panels **B** and **C** have been repeated in *Figure 4—figure supplement 1E–F* for clarity. (**D–J**) Three control (*Shh^{CreER/+};Rosa26^{Ai9/+}*) (**E–G**) and five mutant (*Shh^{CreER/–},Rosa26^{Ai9/+}*) (**H–J**) embryos were induced with Tamoxifen at E8.5 and analyzed for TdTomato (TdT)-expressing *Shh*-descendant cells (in red; **E–F, H–I**) and for Cleaved Caspase-3 expression (in white; **E,G,H,J**) in the larynx at E10.5 (30–34 s). Arrowheads indicate Shh-descendant cells within the epithelium (**I'**) and within the mesenchyme (**H'', I''**). (**K**) The number of TdT-expressing cells found in the mesenchyme in three controls (*Shh^{CreER/+}*) and five mutants (*Shh^{CreER/–}*) was quantified and tested for significance using the Student's *t*-test. Error bars show the standard error of the mean. (**L**) The

*Figure 4 continued on next page*

*Figure 4 continued*

percentage of TdT-expressing cells within the ventral half of the epithelium in three controls ($Shh^{CreER/+}$) and four mutants ($Shh^{CreER/-}$) was quantified and tested for significance using the Student's *t*-test. Source data for panels **K, L** can be found in *Supplementary file 1*. Error bars show the standard error of the mean. *p < 0.05, ***p < 0.0005; n.s. – not significant. Panels **B, C, E, F, G, H, I, J** are single slice images. All other panels are z-projections. D – dorsal; V – ventral (all panels are in the same orientation). All scale bars denote 50 μm.

The online version of this article includes the following figure supplement(s) for figure 4:

**Figure supplement 1.** FOXA2 is downregulated in the early foregut endoderm and absent from the larynx epithelium by E11.5 in $Shh^{-/-}$s.

**Figure supplement 2.** $Ptch^{LacZ}$ and $Gli1^{LacZ}$ expression is present within the larynx epithelium as well as the mesenchyme at E10.5.

**Figure supplement 3.** FOXA2 is not downregulated in the epithelial lamina or the lateral esophageal epithelium during larynx-esophageal separation.

**Figure supplement 4.** Mutant epithelial cells leave the epithelium but do not undergo cell death until E10.5 in the absence of HH signaling.

**Figure supplement 5.** HH signaling is required for the survival of epithelial and mesenchymal cells within the larynx.

**Figure supplement 6.** Loss of *Shh* results in thickening and cell disorganization of the epithelial layer.

---

(*Figure 3—figure supplement 3*), there was an increase in the number of labeled cells within the adjacent mesenchyme surrounding the epithelium in mutant embryos (E9.75–E10.5, *Figure 4H, I, K*, *Figure 4—figure supplement 4D, E*, *Supplementary file 1*) suggesting EMT induction. The low number of these cells suggested that most of the cells leaving the epithelium do not survive. Consistent with this idea, there were high levels of cell death in both the mesenchymal and epithelial tissues of the vocal folds between E9.5 and E11.5, peaking at over 30% of the epithelium (*Figure 4H, J*; *Figure 4—figure supplement 5A–H*). Initially, mesenchymal TdT-expressing cells in $Shh^{CreER/-}$;$Rosa26^{Ai9/+}$s at E9.75 did not express the apoptosis marker Cleaved Caspase-3 (*Figure 4—figure supplement 4F, F'*, *Supplementary file 1*) however by E10.5, the majority of $Shh^{CreER/-}$;$Rosa26^{Ai9/+}$-labeled cells outside the epithelium were apoptotic (*Figure 4H, J*, *Supplementary file 1*). We conclude that most of the vocal fold cells undergoing EMT in $Shh^{-/-}$ embryos are either in the process of undergoing apoptosis or undergo apoptosis shortly after extrusion.

## Initial *Shh*-expressing epithelial cells are replaced by a new cell population in the absence of HH signaling

During the initial period of cell death, proliferation levels remained unchanged. However, by E11.5 there was a significant increase in cell proliferation within the vocal fold epithelium of $Shh^{-/-}$ embryos (*Figure 4—figure supplement 5I–J*). This, and the persistence of a morphologically distinct epithelium, implied that HH-independent mechanisms might contribute to epithelial maintenance. Notably, the $Shh^{-/-}$ epithelium was highly disorganized. Compared to the uniform, one to two cell layers observed in control embryos, mutant embryos had highly variable epithelia containing increased numbers of cell layers (an average of 12 layers; *Figure 4—figure supplement 6A–C*), with an overall thickening of the vocal fold epithelium. This aberrant epithelium continued to persist until at least E13.5, and was composed of rudimentary, poorly keratinized, p63-negative cells that did not recover normal epithelial form or function (*Figure 2—figure supplement 1D, E*).

We asked if the epithelial cells that persist to later stages are descendants of the initial cells marked by *Shh*. Using the same $Shh^{CreER/-}$;$Rosa26^{Ai9/+}$ embryos described above, we found that *Shh* descendants were primarily localized to the ventral half of the vocal fold epithelium by E10.5 (*Figure 4D–F, L*), mirroring previous reports of ventral *Shh* expression in the early foregut (*Burke and Oliver, 2002*; *Moore-Scott and Manley, 2005*; *Motoyama et al., 1998*; *Rankin et al., 2016*; *Sagai et al., 2009*; *Szabó et al., 2009*). As *Shh* is later expressed within the dorsal larynx epithelium at E11.5, this suggests that foregut epithelial cells dynamically regulate *Shh* during these timepoints (*Lungova et al., 2018*; *Lungova et al., 2015*). In marked contrast to control embryos, there was a sharp decrease in *Shh*-descendant cells in $Shh^{CreER/-}$;$Rosa26^{Ai9/+}$ mutants (*Figure 4H, J, L*), indicating that the aberrant epithelium observed at later stages was not descended from the initial epithelium. We then examined the expression of the patterning markers SOX2 and NKX2.1 (*Kim et al., 2019*; *Kuwahara et al., 2020*; *Nasr et al., 2019*; *Que et al., 2007*). At E10.5 there is an absence of NKX2.1 and a significant reduction in nuclear SOX2, which is undetectable by E11.5 (*Figure 5A–E*, *Figure 5—figure supplement 1A–D*). We conclude that larynx epithelial cells are abnormally patterned in $Shh^{-/-}$ embryos.

Loss of *Shh* has been shown to cause an expansion of *Pax1*, a marker of pharyngeal pouches 1–3 in the anterior foregut as well as the expansion of *Foxn1*-expressing thymic progenitors in the third

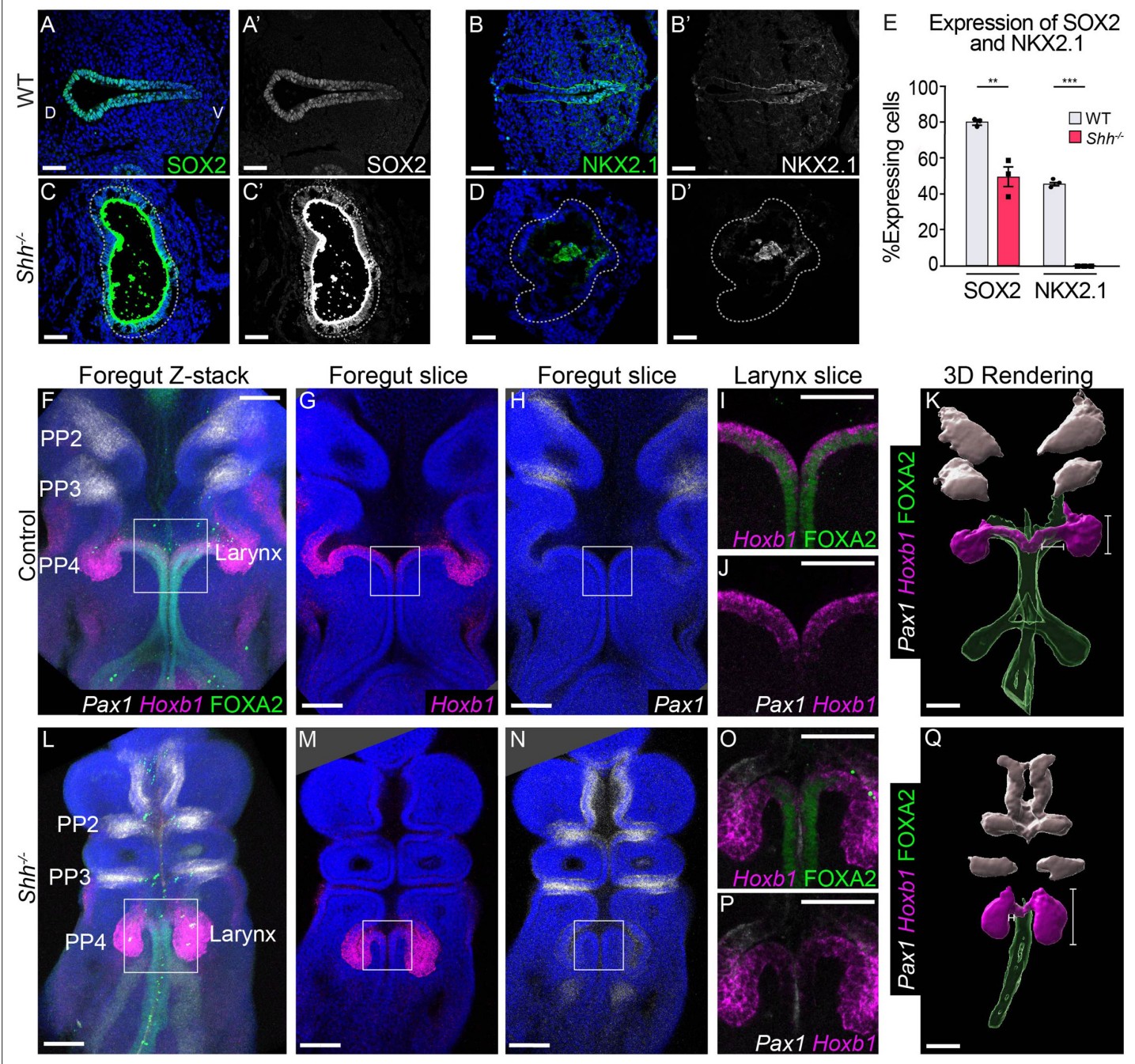

**Figure 5.** *Shh*-descendant larynx epithelial cells are replaced by an unknown population of cells in the absence of HH signaling. SOX2 (**A, C**) and NKX2.1 (**B–D**) expression in three control and *Shh⁻/⁻* larynxes at E10.5 (32–35 s, *n* = 3 per genotype). Apical staining along the epithelium in panels **C–C'**, **D–D'** is non-specific signal from anti-mouse secondary antibodies. (**E**) The percentage of SOX2- and NKX2.1-expressing cells within the epithelium was quantified in three controls and three *Shh⁻/⁻*s each at E10.5 and analyzed for significance using the Student's *t*-test (**p < 0.005 ***p < 0.0005). Error bars plot the standard error of the mean. (**F–J, L–P**) *Pax1* (white), *Hoxb1* (magenta), and FOXA2 (green) expression in three control and *Shh⁻/⁻* foreguts at E10.5 (30–32 s, *n* = 3 per genotype; *Pax1*, *Hoxb1*, and FOXA2 channels were imaged for each replicate). (**K, Q**) Three-dimensional renderings of *Pax1*, *Hoxb1*, and FOXA2 expression domains along the foregut epithelium in control and *Shh⁻/⁻* foreguts at E10.5. Panels **F, L, K, Q** have been repeated in *Figure 5—figure supplement 2* for clarity. Panels **A–D', G–J, M–P** are single slice images. All other panels are z-projections. PP – pharyngeal pouch; D – dorsal; V – ventral (panels **A–D** are in the same orientation). Scale bars denote 50 µm (**A–D**) and 100 µm (**F–M**).

The online version of this article includes the following figure supplement(s) for figure 5:

**Figure supplement 1.** SOX2, NKX2.1, and FOXA2 are absent from the larynx epithelium by E11.5 in *Shh⁻/⁻*s.

**Figure supplement 2.** Pharyngeal pouch marker *Pax1* does not extend into the larynx in the absence of HH signaling.

*Figure 5 continued on next page*

*Figure 5 continued*

**Figure supplement 3.** *Foxn1*-expression is not extended beyond the third pharyngeal pouch in the absence of HH signaling.

pharyngeal pouch (*Johansson et al., 2015*; *Moore-Scott and Manley, 2005*; *Wallin et al., 1996*; *Westerlund, 2013*). To determine if expanded pharyngeal pouch populations replace larynx epithelial cells in *Shh*$^{-/-}$s, we examined the expression of *Pax1*, *Hoxb1* (expressed in the fourth pharyngeal pouches), and *Foxn1* (*Moore-Scott and Manley, 2005*; *Wallin et al., 1996*). We used three-dimensional rendering to examine the spatial distribution of these markers along the foregut (*Figure 5*, *Figure 5—figure supplements 2 and 3*). Consistent with previous findings, *Pax1* was expressed exclusively in pharyngeal pouches 1–3 and was excluded from the epithelium and mesenchyme of the vocal folds in controls (*Figure 5F, H–J*, *Figure 5—figure supplement 2A*; *Johansson et al., 2015*; *Moore-Scott and Manley, 2005*; *Westerlund, 2013*). *Hoxb1*, which was highly expressed in the fourth pharyngeal pouches adjacent to the larynx was also transiently expressed at lower levels within the epithelium of the vocal folds at E10.5, defining it as a marker of the larynx in control and mutant tissues at this timepoint (*Figure 5F, G, J*, *Figure 4—figure supplement 6A", B"* 1–2, *Figure 5—figure supplement 2A–C*). At 30–32 s (E10.0), just prior to widespread cell death within the epithelium, *Shh*$^{-/-}$ embryos had severely altered pharyngeal pouch and larynx morphology. The fourth pharyngeal pouches were contracted toward the larynx in mutants, forming a single continuous structure with the same high *Hoxb1* expression along the lateral walls in anterior sections through the larynx, which resolved into separated pouch and foregut epithelia posteriorly (*Figure 5P–Q*). In some instances, the pouches appeared fused with the anterior larynx with the lateral sides expressing the high levels of *Hoxb1* characteristic of the fourth pharyngeal pouches, and internal domains expressing lower levels of *Hoxb1* consistent with expression in the larynx epithelium (*Figure 4—figure supplement 6A–B*). Despite these severe morphology changes, *Pax1* and *Hoxb1* domains remained unchanged in *Shh*$^{-/-}$ embryos at both E10.75 (37–38 s) and E11.5 (42–44 s) while FOXA2 was either severely reduced or absent from the larynx and anterior foregut (*Figure 5F–Q*, *Figure 5—figure supplement 1E–F*, *Figure 5—figure supplement 2A, D–F*). Similarly, *Foxn1*, which marks the third pharyngeal pouch, did not expand into the larynx (*Figure 5—figure supplement 3C–D*; *Moore-Scott and Manley, 2005*; *Wallin et al., 1996*). This suggests that the abnormally positioned fourth pharyngeal pouches fuse with and replace part of the larynx epithelium in *Shh*$^{-/-}$ embryos while more anterior pouch tissues do not expand into the larynx.

## Discussion

We report a role for HH signaling in regulating the morphogenesis of the presumptive laryngeal epithelium. There is an unexpectedly early role for HH signaling in maintaining the nascent foregut epithelium, which in its absence undergoes an EMT-like process marked by cadherin switching, cell extrusion and ultimately cell death (*Figure 6A*). As this initial population of epithelium dies, it is replaced by an ectopic population of cells likely originating from the fourth pharyngeal pouches (*Figure 6B*). The unexpected presence of this unknown population complicates the previous interpretation of HH mutant phenotypes in the anterior foregut, as changes in gene expression that have been interpreted as reflecting HH-dependent transcriptional changes might instead reflect the properties of this new population of replacement cells.

### The *Shh*$^{-/-}$ larynx epithelium consists of an unknown population of cells with aberrant gene expression

The loss of FOXA2, SOX2, and NKX2.1 (*Figures 4 and 5*, *Figure 5—figure supplements 1 and 2*) could be due to either the transcriptional downregulation of HH-target genes that establish dorsal–ventral patterning or it could reflect the absence of cells that express these foregut markers. The greatly reduced levels of *Shh*-descendant labeling coupled with high levels of cell death (*Figure 4*; *Figure 4—figure supplements 4 and 5*) is consistent with the latter scenario, suggesting that the original *Shh*-expressing endodermal cells are replaced by a secondary population of cells lacking FOXA2. These cells are not part of expanded anterior pharyngeal pouches (pouches 1-3) or thymic domains (*Figure 5*, *Figure 5—figure supplements 1 and 2*; *Gordon et al., 2001*; *Moore-Scott and Manley, 2005*; *Wallin et al., 1996*). The high levels of *Hoxb1* expression in the *Shh*$^{-/-}$ epithelium

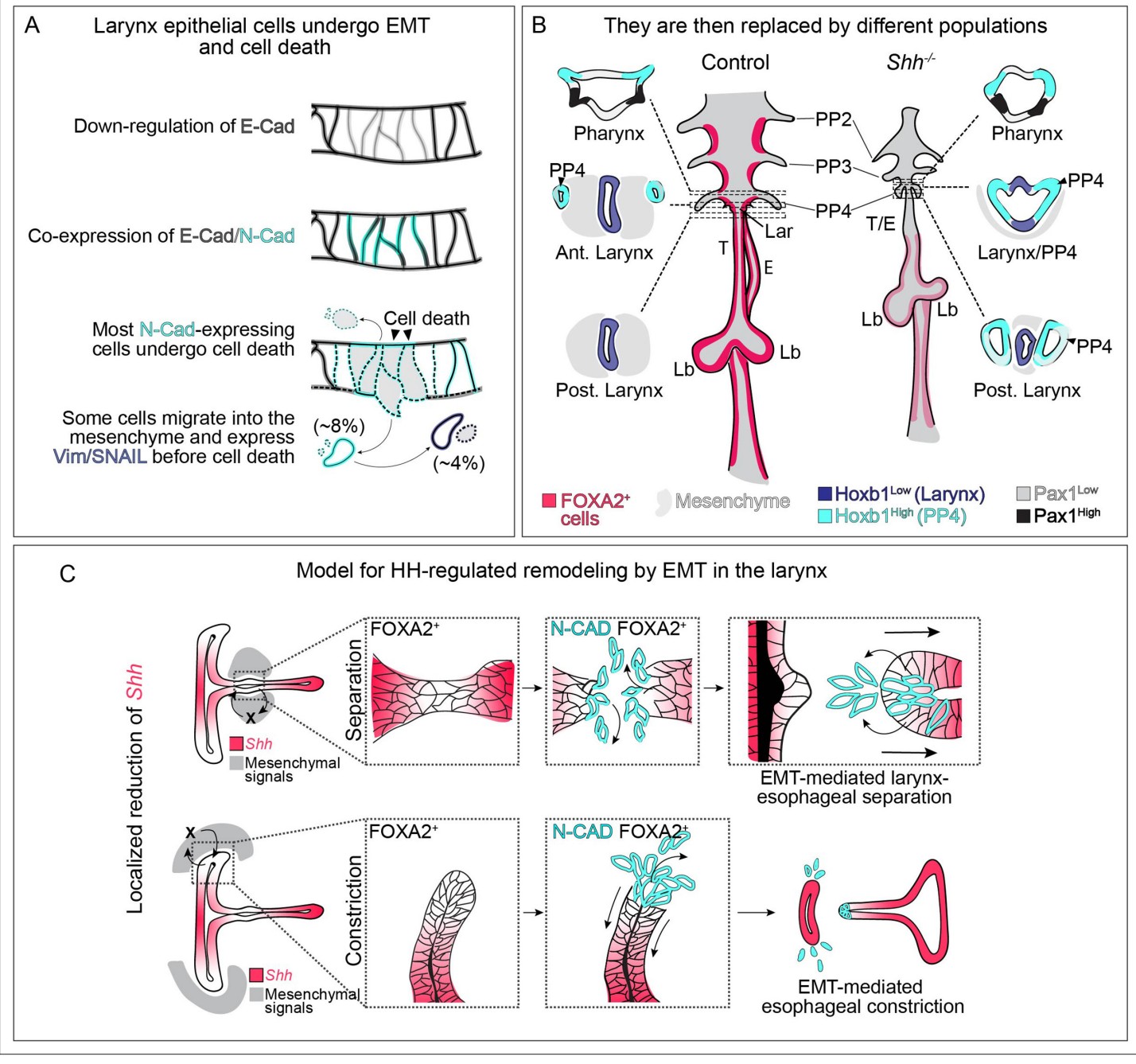

**Figure 6.** Dynamic HH signaling drives the homeostasis of the early anterior foregut endoderm and may also regulate later stages of larynx remodeling. (**A**) Epithelial cells marked by E-Cadherin (gray) in the anterior foregut undergo a cadherin switch (marked by the induction of N-Cadherin expression in cyan), cell death and are cleared from the epithelium in the absence of HH signaling. Some cells (8%) migrate out of the foregut epithelium and a subset express mesenchymal markers (Vimentin and SNAIL expression in dark blue; 4%) prior to cell death. (**B**) We propose that the original larynx epithelial cells (expressing FOXA2 in pink) are replaced by cells derived from the fourth pharyngeal pouch (marked by high *Hoxb1* expression in cyan) which have fused to the laryngeal epithelium (marked by low *Hoxb1* expression in dark blue) in anterior sections. (**C**) Proposed model for epithelial-to-mesenchymal transition (EMT)-mediated morphogenesis at later stages of larynx development. Dynamic *Shh* (pink) expression within the larynx epithelium drives EMT-mediated morphogenesis resulting in larynx-esophageal separation and esophageal constriction. PP 2/3/4 – pharyngeal pouches #2–4, Lar – larynx, T – trachea, E – esophagus, Lb – lung buds.

suggest that adjacent fourth pharyngeal pouch tissues contribute to the mutant larynx in the anterior foregut (*Figure 5P–Q*, *Figure 4—figure supplement 6A–B*). This fusion of the two compartments is likely due to high levels of cell death within both the vocal fold mesenchyme as well as throughout the adjacent pharyngeal arch mesenchyme (arches 2 and 3) (*Moore-Scott and Manley, 2005*), which we speculate result in the collapse of the fourth pharyngeal pouches toward the midline of the foregut (*Figure 4—figure supplement 5H*, *Figure 6B*). It is presently unclear how extensively this population contributes to the replacement of *Shh* descendants throughout the anterior foregut. FOXA2 expression is also absent from the more posterior trachea and esophagus compared to controls (*Figure 5—figure supplement 1F*). This suggests that FOXA2 regulation by *Shh* extends throughout the anterior foregut and that posterior tracheal/esophageal tissues might also be replaced in *Shh*$^{-/-}$s. However as this tissue does not express *Hoxb1* it is unlikely to be formed from the fourth pharyngeal pouches.

## HH-dependent epithelial changes are mediated by a partial EMT

Several lines of evidence indicate that larynx epithelial cells undergo a partial EMT in *Shh*$^{-/-}$ embryos. First, they undergo the switch from E-Cadherin to N-Cadherin described above. Subsequently, some of these cells are found in the mesenchyme while others are present in the lumen or appear in the process of extrusion (*Figure 2I–K*; *Figure 4I, K*). Some of the cells within the mesenchyme express Vimentin and SNAIL (*Figure 3—figure supplement 3*), markers indicative of mesenchymal cell types that are not expressed in the original epithelial cells (*Barrallo-Gimeno and Nieto, 2005*; *Casas et al., 2011*; *Huang et al., 2012*; *Jägle et al., 2017*; *Liu et al., 2015*; *Mendez et al., 2010*; *Vuoriluoto et al., 2011*). This suggests that epithelial cells are capable of becoming mesenchymal cells, although they subsequently undergo apoptosis. A characterization of the identity of these presumably short-lived transitioning cells remains elusive, as the best-described mesenchymal markers, *Foxf1* and *Sox9* are significantly reduced in *Shh*$^{-/-}$ mesenchyme and therefore cannot be used to determine if the cells adopt a larynx mesenchymal fate (*Nasr et al., 2021*; *Nasr et al., 2019*). It is also important to emphasize that even prior to the onset of cell death, only about 8% of cells are found in the mesenchyme, and only half of these cells express Vimentin/SNAIL (9/21 cells in the mesenchyme at E9.75) (*Figures 2 and 4*, *Figure 4—figure supplement 5*). All mutant epithelial cells undergo cadherin switching and lose cell stratification (*Figure 2*, *Figure 2—figure supplement 1*). The majority of these cells do not leave the epithelium and do not express SNAIL or Vimentin while undergoing apoptosis. It remains unclear how these cells are removed from the epithelium. Possibilities include an EMT-independent destabilization of the epithelial layer due to extensive cell death, a loss of epithelial contacts to the basement membrane (signaled by loss of Laminin and p63), or cell death-induced cell extrusion (*Kuipers et al., 2014*; *Ohsawa et al., 2018*; *Rosenblatt et al., 2001*).

   In contrast to the phenotypes observed in germline mutants, the changes in cell properties of low-*Shh*-expressing cells in the wild-type remodeling larynx during larynx-esophageal separation were comparatively mild. These cells do not undergo apoptosis, perhaps because of the presence of low-level HH signaling. Most prominently, they do not downregulate E-Cadherin as they do in *Shh*$^{-/-}$ larynx epithelia (*Figures 1 and 2*, *Figure 1—figure supplement 1*). One explanation for the milder phenotype likely lies in their differential regulation of FOXA2, which is downregulated in *Shh*$^{-/-}$ foreguts but is maintained during larynx-esophageal separation (*Figure 4—figure supplements 1 and 3*). *Cdh1* (encoding E-Cadherin) has been reported to be transcriptionally regulated by FOXA2 in gastrulating endoderm as well as oral and breast cancer cells (*Bow et al., 2020*; *Scheibner et al., 2021*; *Zhang et al., 2015*). Thus, the reduction of FOXA2 from the endoderm in *Shh*$^{-/-}$s could initiate changes in cell adhesion resulting from reduced production of E-Cadherin. Both cell populations upregulate N-Cadherin (*Figure 6A, C*), suggesting that it is negatively regulated by *Shh* in both contexts. The coexpression of E-Cadherin and N-Cadherin during laryngeal remodeling has also been observed in FOXA2-expressing gastrulating endodermal progenitors during EMT (*Scheibner et al., 2021*).

## Does HH signaling directly regulate larynx epithelial genes?

Prior to being lost from the foregut, FOXA2 expression is gradually reduced in *Shh*$^{-/-}$ foreguts (*Figure 4A–C*, *Figure 4—figure supplement 1D, F, H*; *Yamagishi et al., 2003*). *FoxA2* is a direct transcriptional target of SHH in the neural tube (*Oosterveen et al., 2012*; *Peterson et al., 2012*; *Sasaki and Hogan, 1994*) and the HH-target genes *Ptch1* and *Gli1* are expressed within the foregut epithelium (*Figure 4—figure supplement 2A, C*). This is consistent with the possibility that HH signaling

could regulate *FoxA2* or perhaps other epithelial genes through autocrine signaling (*Motoyama et al., 1998*; *Yamagishi et al., 2003*). However, as has been previously noted in other foregut tissues, the expression of HH-responsive target genes is much lower in the epithelium than in the mesenchyme (*Figure 4—figure supplement 2C*; *Han et al., 2017*; *Moore-Scott and Manley, 2005*; *Motoyama et al., 1998*; *Rankin et al., 2016*). Thus the most plausible scenario is that HH signaling indirectly regulates epithelial fate through paracrine signaling to the adjacent mesenchyme (*Han et al., 2020*; *Han et al., 2017*; *Nasr et al., 2021*; *Rankin et al., 2016*; *Yang et al., 2021*). Consistent with the latter possibility, there is widespread cell death in the *Shh⁻/⁻* mesenchyme that precedes that in the epithelium (*Figure 4*, *Figure 4—figure supplement 5G, H*). This results in dramatic changes to the composition of the mesenchyme, including the upregulation of multiple TGFβ family members that have well-established roles in inducing EMT as well as antagonizing HH signaling during thymic/parathyroid and pancreas induction (*Hebrok et al., 1998*; *Katsuno et al., 2013*; *Kim and Hebrok, 2001*; *Mercado-Pimentel and Runyan, 2007*; *Moore-Scott and Manley, 2005*; *Nawshad et al., 2004*; *Schnaper et al., 2003*; *Thiery et al., 2009*).

Confirming previous studies, we find that *Shh* is dynamically expressed during larynx development (*Lungova et al., 2018*; *Lungova et al., 2015*; *Sagai et al., 2009*). In addition to its downregulation from regions of the larynx epithelium that express N-Cadherin, the relative levels of *Shh* within *Shh*-expressing domains of the epithelium are highly dynamic at later stages, where there is an overall reduction in *Shh* within the dorsal half of the larynx epithelium, which is contiguous with the esophagus, compared to the ventral half (*Figure 1B, H*). Additionally, lower levels of *Shh* have been reported in the trachea compared to the esophageal epithelium at later stages (*Nasr et al., 2021*). *Shh* expression in the larynx is regulated by three distinct enhancers that occupy largely non-overlapping regions of activity along the dorsal–ventral axis of the larynx (*Sagai et al., 2017*; *Sagai et al., 2009*; *Tsukiji et al., 2014*). While it remains unclear how they are regulated, differential enhancer utilization is a plausible mechanism for regional regulation of *Shh* along the foregut.

## A global role for hedgehog signaling in anterior foregut organogenesis

We propose that regionalized reduction in *Shh* within the anterior foregut triggers partial EMT as a key step in driving the morphogenesis of other foregut-derived organs. Alternatively, there may be additional regional factors that are required to activate partial EMT upon withdrawal of HH signaling. HH is locally restricted along the foregut endoderm at the initiation sites of multiple foregut-derived organs including the thymus, the pancreas, the thyroid, and the liver (*Apelqvist et al., 1997*; *Bain et al., 2016*; *Bort et al., 2006*; *Fagman et al., 2004*; *Gordon and Manley, 2011*; *Hebrok, 2000*; *Hebrok et al., 1998*; *Moore-Scott and Manley, 2005*; *Westerlund, 2013*). It is unclear why HH restriction is required in these different contexts and if they share a common mechanism such as EMT. The liver bud is generated from foregut tissue that lacks *Shh* expression and subsequently undergoes EMT into the adjacent mesenchyme (*Bort et al., 2006*; *Mu et al., 2020*). Additionally, loss of *Shh* and the expression of N-Cadherin within the foregut epithelium mark the site of the presumptive dorsal and ventral pancreatic buds, though N-Cadherin is dispensable for the initial stages of pancreatic budding (*Esni et al., 2001*; *Johansson et al., 2010*). While the role of *Shh* has not been directly studied in this process, *Hhex* mutants, which fail to undergo EMT of the liver bud also mis-express *Shh* in the epithelium (*Bort et al., 2006*). Given the role for FOXA2 in regulating EMT in gastrulating endoderm, HH signaling could act either directly or indirectly to maintain FOXA2 (*Scheibner et al., 2021*). This could include the maintenance of FOXA2 expression/activity or co-regulation of a set of common downstream targets.

# Materials and methods

**Key resources table**

| Reagent type (species) or resource | Designation | Source or reference | Identifiers | Additional information |
|---|---|---|---|---|
| Genetic reagent (*M. musculus*) | *Shh^tm1amc* (*Shh^+/−*) | Jackson Laboratory (*Lewis et al., 2001*) | Jackson Cat# 003318, MGI Cat# 3584154, RRID:MGI:3584154 | Swiss Webster background |

*Continued on next page*

*Continued*

| Reagent type (species) or resource | Designation | Source or reference | Identifiers | Additional information |
|---|---|---|---|---|
| Genetic reagent (*M. musculus*) | *Cg-Shh*$^{tm1(EGFP/cre)Cjt}$ (*Shh*$^{GFP-Cre}$) | Jackson Laboratory (*Harfe et al., 2004*) | Jackson Cat#: 005622; MGI Cat#: 92505; RRID: IMSR_JAX:005622 | Mixed background |
| Genetic reagent (*M. musculus*) | *Ptch1*$^{tm1Mps}$/J (*Ptch*$^{LacZ}$) | Jackson Laboratory (*Goodrich et al., 1997*) | Jackson Cat#: 003081; MGI Cat#: 42441; RRID: IMSR_JAX:003081 | BL6-background |
| Genetic reagent (*M. musculus*) | *Gli1*$^{tm2Alj}$/J (*Gli1*$^{LacZ}$) | Laboratory of Dr. Aaron Zorn | Jackson Cat#: 008211; MGI Cat#: 2449767; RRID: IMSR_JAX: 008211 | |
| Genetic reagent (*M. musculus*) | *B6.129S6-Shh*$^{tm2(cre/ERT2)Cjt}$/J (*Shh*$^{CreER/+}$) | Laboratory of Dr. Susan Mackem (*Harfe et al., 2004*) | Jackson Cat#: 005623 MGI 92504 RRID: IMSR_JAX:005623 | Swiss Webster background |
| Genetic reagent (*M. musculus*) | *B6.Cg-Gt(ROSA)26Sor*$^{tm9(CAG-tdTomato)Hze}$/J (*Rosa26*$^{Ai9/+}$) | Jackson Laboratory | Jackson Cat#: 007909 MGI: 155793 RRID: IMSR_JAX: 007909 | BL6-background |
| Genetic reagent (*M. musculus*) | Swiss Webster Wildtype | Charles River | Charles River Cat# NCI 551 IMSR Cat# TAC:sw, RRID:IMSR_TAC:sw | |
| Chemical compound, drug | Tamoxifen | Sigma-Aldrich | Cat#: T5648-1G Lot/batch#: WXBD2299V | |
| Other | Trizol | Life Technologies | Cat#: 10296010 | Used for RNA extraction |
| Commercial assay, kit | NEBNext Ultra II Directional RNA library prep kit | New England Biolabs | Cat#: E7760L | |
| Commercial assay, kit | In Situ Cell death detection Kit | Roche | Cat# 12156792910 Lot#: 11520500 | OCT and paraffin |
| Other | 4′,6-Diamidino-2-phenylindole (DAPI) | Invitrogen | Cat# D1306 Lot#: 2208529 | Nuclear stain for OCT and paraffin sections and whole mounts (1:5000) |
| Other | Prolong Gold Antifade | Thermo Fisher Scientific | Cat#: P36930 | Used to mount OCT/paraffin section stains |
| Chemical compound, drug | Histodenz | Sigma-Aldrich | Cat#: D2158-100G Lot #: WXBD3838V | Used to clear whole-mount embryos |
| Chemical compound, drug | *N*-Methyl-acetimide | Sigma-Aldrich | Cat#: M26305-100G | Used to clear whole-mount embryos |
| Other | Low-melt agarose | Sigma-Aldrich | Cat#: A2576-5G Lot#: SLCG3476 | Used at 1.5% agarose in water (wt/vol) |
| Antibody | SOX9 (rabbit polyclonal) | Millipore | Cat#: AB5535; Lot#: 3389351 RRID:AB_2239761 | OCT and paraffin (1:200) |
| Antibody | SOX2 (mouse monoclonal) | Santa-Cruz | Cat#: sc-365823; LOT#: E1619 RRID:AB_10842165 | OCT and paraffin (1:200) |
| Antibody | NKX2.1 (TTF-1) (mouse monoclonal) | Santa-Cruz | Cat#: sc-53136 LOT#: B2219 RRID:AB_793529 | OCT (1:200) |
| Antibody | FOXA2 (rabbit monoclonal) | Abcam | Cat#: ab108422; LOT#: GR3289185 RRID:AB_11157157 | OCT and paraffin (1:300) |
| Antibody | FOXA2 (mouse monoclonal) | DSHB | Cat#: 4c7 RRID:AB_528255 | OCT (1:50) |
| Antibody | N-Cadherin (rabbit monoclonal) | Cell Signaling Technologies | Cat#: 13116S RRID:AB_2687616 | OCT (1:200) |

*Continued on next page*

*Continued*

| Reagent type (species) or resource | Designation | Source or reference | Identifiers | Additional information |
|---|---|---|---|---|
| Antibody | E-Cadherin (rabbit monoclonal) | Cell Signaling Technologies | Cat#: 3195S RRID:AB_2291471 | OCT and paraffin (1:200) |
| Antibody | Vimentin (rabbit monoclonal) | Cell Signaling Technologies | Cat#: 5741T RRID:AB_10695459 | OCT (1:200) |
| Antibody | P63 (mouse monoclonal) | Abcam | Cat#: ab735 RRID:AB_305870 | OCT and paraffin (1:200) |
| Antibody | Cleaved-Caspase3 (D165) (rabbit monoclonal) | Cell Signaling Technologies | Cat#: 9664S RRID:AB_2070042 | OCT and paraffin (1:200) |
| Antibody | GFP (chicken polyclonal) | Aves | Cat#: 1020; LOT#: 1229FP08 RRID:AB_10000240 | OCT (1:500) |
| Antibody | RFP (Rabbit polyclonal) | Rockland | Cat#: 600-401-379; LOT#: 46317 RRID:AB_2209751 | OCT (1:100) |
| Antibody | ECAD-488 (24E10) (rabbit monoclonal) | Cell Signaling Technologies | Cat#: 3199S RRID:AB_10691457 | OCT (1:200) |
| Antibody | Phospho-Histone H3 (rabbit polyclonal) | Millipore | Cat#: 06-570; LOT#: 2972863 RRID:AB_310177 | OCT and paraffin (1:200) |
| Antibody | Rabbit Isotype Control (DA1E) (rabbit monoclonal) | Cell Signaling Technologies | Cat#: 3900S RRID:AB_1550038 | OCT (4 µg/ml) |
| Antibody | RAB-11 (rabbit monoclonal) | Cell Signaling Technologies | Cat#: 5589T RRID:AB_10693925 | OCT (1:100) |
| Antibody | Beta-Catenin (rabbit polyclonal) | Thermo Fisher Scientific | Cat#: 71-2700 RRID:AB_2533982 | OCT (1:500) |
| Antibody | Laminin (rabbit polyclonal) | Sigma-Aldrich | Cat#: L9393-100UL; LOT#: 099M4886V RRID:AB_477163 | Paraffin (1:100) |
| Antibody | anti-Rabbit Alexa 488 (goatpolyclonal) | Thermo Fisher Scientific | Cat#: A11034 RRID:AB_2576217 | OCT and paraffin (1:250) |
| Antibody | anti-Mouse Alexa 568 (goat polyclonal) | Thermo Fisher Scientific | Cat#: A11004 RRID:AB_2534072 | OCT and paraffin (1:250) |
| Antibody | anti-Chicken Alexa 568 (goat polyclonal) | Thermo Fisher Scientific | Cat#: A11041 RRID:AB_2534098 | OCT (1:250) |
| Antibody | anti-Rabbit Alexa 647 (goat polyclonal) | Life Technologies | Cat#: A27040 RRID:AB_2536101 | OCT (1:250) |
| Commercial assay, kit | V3.0 HCR RNA-FISH Kit Probe Hybridization buffer | Molecular Instruments | LOT#: BPH02324 | Whole-mount embryos |
| Commercial assay, kit | V3.0 HCR RNA-FISH Kit Probe Wash buffer | Molecular Instruments | LOT#: BPW02123 | Whole-mount embryos |
| Commercial assay, kit | V3.0 HCR RNA-FISH Kit Amplification buffer | Molecular Instruments | LOT#: BAM01923 | Whole-mount embryos |
| Commercial assay, kit | Cdh1 HCR probe (*M. musculus*) | Molecular Instruments | Probe Lot#: PRI679 MGI Accession: 12550 | Hairpin-B2-Alexa488 |
| Commercial assay, kit | Cdh2 HCR probe (*M. musculus*) | Molecular Instruments | Probe Lot#: PRH832 MGI Accession: BC022107 | Hairpin-B1- Alexa594 |
| Commercial assay, kit | Shh HCR probe (*M. musculus*) | Molecular Instruments | Probe Lot#: PRA909 MGI Accession: NM_009170 | Hairpin-B1-Alexa594 |

*Continued on next page*

*Continued*

| Reagent type (species) or resource | Designation | Source or reference | Identifiers | Additional information |
|---|---|---|---|---|
| Commercial assay, kit | Pax1 HCR probe (*M. musculus*) | Molecular Instruments | Probe Lot#: PRH830 MGI Accession: NM_008780.2 | Hairpin-B1-Alexa594 |
| Commercial assay, kit | Hoxb1 HCR probe (*M. musculus*) | Molecular Instruments | Probe Lot#: PRE343 | Hairpin-B4-Alexa647 |
| Commercial assay, kit | Foxn1 HCR probe (*M. musculus*) | Molecular Instruments | Probe Lot#: PRN998 | Hairpin-B4-Alexa647 |
| Software, algorithm | HISAT2 v2.1.0 | *Pertea et al., 2016*; *Pertea et al., 2015* | RRID:SCR_015530 | Used for RNA-seq analysis |
| Software, algorithm | StringTie v1.3.6 | *Pertea et al., 2016*; *Pertea et al., 2015* | RRID:SCR_016323 | Used for RNA-seq analysis |
| Software, algorithm | Imaris v9.9.1 software | Bitplane Inc | RRID:SCR_007370 | Used for 3D rendering of surfaces |

## Embryonic manipulations

All experiments involving mice were approved by the Institutional Animal Care and Use Committee at the University of Texas at Austin (protocol AUP-2019-00233). The $Shh^{tm1amc}$ null allele (referred to as $Shh^{+/-}$) was maintained on a Swiss Webster background (*Lewis et al., 2001*). The $Cg$-$Shh^{tm1(EGFP/cre)Cjt}$ ($Shh^{GFP-Cre}$) (*Harfe et al., 2004*), the $Ptch1^{tm1Mps}$/J ($Ptch^{LacZ}$) (*Goodrich et al., 1997*), the $Gli1^{tm2Alj}$/J ($Gli1^{LacZ}$) (*Bai et al., 2002*), and the $Shh^{CreER/+}$;$Rosa26^{Ai9/+}$ lines (*Harfe et al., 2004*; *Srinivas et al., 2001*) were maintained on mixed genetic backgrounds. To label *Shh*-descendant cells, pregnant mice containing $Shh^{CreER}$;$Rosa26^{Ai9}$ embryos were injected intraperitoneally with 3 mg of Tamoxifen (Sigma Aldrich, T5648-1G) per 40 g.

## Gene expression

RNA was extracted using Trizol reagent (Life Technologies, 10296010) and DNAse treated. For bulk RNA-seq, vocal fold tissue was dissected from two sets of 3-pooled control and $Shh^{-/-}$ embryos at E10.5 (32–35 s). Libraries were generated using the NEBNext Ultra II Directional RNA library prep kit and single-end sequenced on the Illumina NextSeq 500 platform at a depth of ~40,000,000 reads/sample. Sequenced reads were aligned to the mm10 genome using HISAT2 v2.1.0 and assembled into genes using StringTie v1.3.6 (*Pertea et al., 2016*; *Pertea et al., 2015*). The RNA-seq is accessible from GEO (accession number GSE190281) and differentially expressed genes are listed in *Figure 2—source data 1*.

## Immunofluorescence

All immunofluorescence replicates (denoted by *n*) refer to independent biological replicates from different embryos.

For paraffin embedding, embryos were fixed overnight in 10% formalin, sectioned to 5 µm and incubated in three 5-min washes of boiling 10 mM sodium citrate buffer, pH 6.0 prior to antibody incubation. For cryosection embedding, embryos were fixed for 1 hr in 4% paraformaldehyde at room temperature, sucrose protected, embedded in OCT (optimal cutting temperature compound) and sectioned to 10 µm. Samples were then permeabilized in 0.06% PBST (phospho-buffered saline with 0.06% Triton-X) prior to blocking. Paraffin and OCT sections were blocked in 3% bovine serum albumin and 5% normal goat serum/PBST (0.1% Tween-20) for 1 hr at room temperature and, following an overnight incubation in primary antibody at 4°C (see Key Resources Table for a list of all antibodies), incubated in secondary antibodies for 1 hr at room temperature. Apoptosis was detected on OCT-embedded sections by TUNEL staining, using the In Situ Cell death detection Kit (Roche, 12156792910). All samples were counterstained in 4',6-diamidino-2-phenylindole (DAPI; Invitrogen, D1306) for 10 min at room temperature and mounted in ProLong Gold Antifade (Thermo Fisher Scientific, P36930) prior to imaging. The E-Cadherin-488, N-Cadherin, and TdT triple stains (*Figure 3*) were imaged on a Nikon Eclipse Ti-2 microscope equipped with a 60x, 1.40NA objective; a Visitech iSIM super-resolution confocal scan head; and a Photometrics Kinetix22 camera. All other images were obtained using a Zeiss LSM 710/Elyra S.1 confocal microscope and 10x, 20x, or 63x objectives.

To visualize E-cadherin and N-Cadherin coexpressing cells within the larynx epithelium, OCT-embedded sections were permeabilized, blocked, and incubated in unconjugated N-Cadherin/ goat anti-rabbit Alexa 647 as specified above. Sections were then blocked in Rabbit IgG isotype control (Cell Signaling Technologies, 3900S) (in 5% normal goat serum, 1% Triton-X, PBS) for 1 hr at room temperature. Following an overnight incubation in E-Cadherin-488 at 4°C, samples were washed in 1× PBS, counterstained with DAPI as described above, and mounted in ProLong Gold Antifade. For whole-mount immunofluorescent staining, embryos were processed as described by *Nasr et al., 2019*. To image, whole-mount stained embryos were embedded in 1.5% low-melt agarose (Sigma, A5030) cooled to room temperature, and cleared overnight using Ce3D++ which was prepared with a high concentration of iohexol as described by *Anderson et al., 2020*.

E-Cadherin localization along the apical–basal axis of the epithelium was measured in Fiji using the average fluorescent intensity of E-Cadherin (normalized to background) within a selected region along the lateral wall of the vocal folds, divided into six equal regions from the apical to the basal end of the epithelium. Relative levels of RAB-11, GFP, FOXA2, *Shh*, and *Cdh1* along the larynx epithelium was measured in Fiji using a 25–35- or 50-pt-thick line scan that was normalized to background fluorescence.

## Whole-mount fluorescent in situ hybridization (HCR)

All whole-mount fluorescent in situ hybridization replicates (denoted by *n*) refer to independent biological replicates from different embryos.

Whole-mount HCR was carried out on whole embryos or cultured larynx explants as previously described in *Anderson et al., 2020*. Briefly, samples were digested in Proteinase K (E9.5 embryos for 10 minutes, E10.5 embryos for 18 minutes, and E11.5 embryos for 30 minutes), incubated in 4 nM probe overnight at 37°C, and then in 60 pmol hairpin per 0.5 ml of amplification buffer (Molecular Instruments) overnight at room temperature. After incubation with the hairpins, samples were washed and counterstained in DAPI overnight as specified by *Anderson et al., 2020*. Samples were then embedded in low-melt agarose and cleared in CeD3++ as described in *Anderson et al., 2020* before imaging. See Key Resources Table for list of HCR probes used in the study.

## Whole-mount HCR coupled with immunofluorescence

Whole-mount HCR was carried out on whole embryos or dissected trunk tissue as described above with the following changes. Samples were digested in Proteinase K for half the normal HCR digestion time, incubated in 4 nM probe overnight at 37°C, and then in 60 pmol hairpin per 0.5 ml of amplification buffer (Molecular Instruments) overnight at room temperature. After incubation with the hairpins, samples were washed 2× in PBS and 1× in PBST, blocked in 5% normal goat serum/PBT (0.2% Triton X-100) for 2 hr at room temperature, and incubated in primary antibody (in block) overnight at 4°C. Following primary antibody incubation, embryos were washed 5× in PBS (1 hr per wash) at room temperature and incubated in secondary antibody (in block) overnight at 4°C. Following secondary antibody incubation samples were washes 3× in PBS (20 min per wash), counterstained in DAPI overnight and embedded and cleared in low-melt agarose and CeD3++, respectively, as described in *Anderson et al., 2020* before imaging. Samples were imaged on a Nikon W1 spinning disk confocal equipped with dual monochromatic Andor EMCCD cameras (10x and 20x objectives).

## Three-dimensional rendering on Imaris

HCR-coupled immunofluorescent whole mounts were imaged as mentioned above and processed using the Surface rendering tool on Imaris 9.9.1. The 3D surfaces for each channel imaged were generated using the same fluorescent intensity ranges across control and mutant samples, with a smoothing pixel size of 4 µm. Surfaces were false colored as separated objects, and any surfaces generated from auto-fluorescent blood cells (which were defined as cell clusters present in all fluorescent channels) were manually deleted from the surface rendering after generation.

## Acknowledgements

We thank John Wallingford and Dan Dickinson for comments on the paper. We thank Dan Dickinson and Naomi Stolpner for use of the Nikon Eclipse microscope. We thank John Wallingford for use of the Zeiss LSM confocal microscope. We thank Susan Mackem for providing the *Shh*^CreER line. We thank

Matt Anderson for assistance with HCR. This work was supported by NIH R01 HD090163 (to SAV and HJ), NIH R01 HD093363 (to AMZ), NIH F30 HL142201 (to TN), a Continuing Graduate Fellowship and Provost's Graduate Excellence Fellowship (to JR), a TIDES Summer Fellowship (to AEB), and an Experiential Learning Summer Scholarship (to ERY).

## Additional information

### Funding

| Funder | Grant reference number | Author |
|---|---|---|
| National Institutes of Health | RO1 HD090163 | Hongkai Ji |
| National Institutes of Health | RO1 HD093363 | Aaron M Zorn |
| National Institutes of Health | F30 HL142201 | Talia Nasr |
| University of Texas at Austin | Continuing Graduate Fellowship | Janani Ramachandran |
| University of Texas at Austin | Provost's Graduate Excellence Fellowship | Janani Ramachandran |
| University of Texas at Austin | TIDES Summer Fellowship | Anna E Bardenhagen |
| University of Texas at Austin | Experiential Learning Summer Scholarship | Ellen R Yates |

The funders had no role in study design, data collection, and interpretation, or the decision to submit the work for publication.

### Author contributions

Janani Ramachandran, Conceptualization, Data curation, Formal analysis, Validation, Investigation, Visualization, Writing – original draft, Writing – review and editing; Weiqiang Zhou, Data curation, Formal analysis; Anna E Bardenhagen, Talia Nasr, Investigation, Writing – review and editing; Ellen R Yates, Data curation, Investigation, Writing – review and editing; Aaron M Zorn, Supervision, Funding acquisition, Methodology, Writing – review and editing; Hongkai Ji, Conceptualization, Supervision, Funding acquisition, Investigation, Methodology, Writing – original draft, Project administration, Writing – review and editing; Steven A Vokes, Conceptualization, Data curation, Supervision, Funding acquisition, Investigation, Writing – original draft, Project administration, Writing – review and editing

### Author ORCIDs

Janani Ramachandran ![ORCID] http://orcid.org/0000-0003-1231-1749
Talia Nasr ![ORCID] http://orcid.org/0000-0002-2473-5402
Aaron M Zorn ![ORCID] http://orcid.org/0000-0003-3217-3590
Steven A Vokes ![ORCID] http://orcid.org/0000-0002-1724-0102

### Ethics

All experiments involving mice were approved by the Institutional Animal Care and Use Committee at the University of Texas at Austin (protocol AUP-2019-00233).

### Decision letter and Author response

Decision letter https://doi.org/10.7554/eLife.77055.sa1
Author response https://doi.org/10.7554/eLife.77055.sa2

## Additional files

### Supplementary files
• Transparent reporting form

• Supplementary file 1. Number of TdT-expressing and Cleaved Caspase-3-expressing cells in the larynx epithelium and mesenchyme of control (Shh^CreER/+;Rosa26^Ai9/+) and mutant (Shh^CreER/−;Rosa26^Ai9/+) embryos at E9.75 (26–29 s) and E10.5 (31–35 s). Cell numbers were quantified from the ventral half of the epithelium in each section analyzed. This table provides source data for figure panels *Figure 4K–L* and *Figure 4—figure supplement 4A–F*. Ventral epi. – ventral epithelium.

## Data availability

Sequencing data have been deposited in GEO under accession code GSE190281.

The following dataset was generated:

| Author(s) | Year | Dataset title | Dataset URL | Database and Identifier |
|---|---|---|---|---|
| Ramachandran J, Zhou W, Bardenhagen AE, Ji H, Vokes SA | 2021 | Hedgehog signaling is essential to maintain epithelial identity during larynx and foregut morphogenesis | https://www.ncbi.nlm.nih.gov/geo/query/acc.cgi=GSE190281 | NCBI Gene Expression Omnibus, GSE190281 |

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
