## [Editor Report]

The authors present cellular and genetic data, combining mutant analysis and RNA-sequencing, that together support a functional role for Shh in repressing the epithelial-to-mesenchymal transition (EMT) in the developing larynx during larynx-esophageal separation. In the absence of Shh, cells undergo EMT and are replaced with a novel epithelial cell population of unknown origin. These results make a significant contribution to the field by illuminating how the larynx develops.

---

## [Decision Letter]

**Decision letter after peer review:**

Thank you for submitting your article "Hedgehog regulation of epithelial cell state and morphogenesis in the larynx" for consideration by *eLife*. Your article has been reviewed by 3 peer reviewers, and the evaluation has been overseen by a Reviewing Editor and Marianne Bronner as the Senior Editor. The following individual involved in the review of your submission has agreed to reveal their identity: Nancy Manley (Reviewer #2).

The reviewers have discussed their reviews with one another. They found the work to be potentially very interesting but they also raised numerous issues that would need to be addressed. If you can address their concerns, we would be happy to receive an appropriately revised version of the manuscript. Please pay close attention to the detailed reviews below.

*Reviewer #1 (Recommendations for the authors):*

That being said, a few issues need to be addressed.

1. The authors emphasize EMT in foregut development. However, the evidence for EMT needs further supports. In Figure 2E, F why the mesenchyme is negative for N-cad, while the mesenchyme in 3P,Q is positive? In this reviewer's opinion, it is possible that deletion of Shh reduces the levels of matrix proteins that attach the epithelium to the basement, and the mutant epithelial cells become loosely attached. During the dynamic morphogenesis process these mutant cells are pushed away, not necessarily going through EMT. Once detached from the BM, these epithelial cells cannot survive and go through apoptosis. This could be the reason for so many cell deaths in the mutants.

2. The authors suggested that Pax1+ cells expand in place of ShH^+^ cell derivatives. Although the lineage tracing in Figure 4E-H seems interesting, it is known that E10.5 foregut of Shh null mutants is severely disrupted. Is there any way to confirm that the section is taken at the same level as the wildtype control? From proximal (the larynx) to distal (stomach) shh expression patten is dynamically changed. In addition, the staining for cleaved caspase 3 seems not very specific.

3. Similar to the issue raised in 2, is there any way to make sure that the mutant and wildtype tissue sections shown in Figure 4—figure supplement 4B are taken at the same level? The cellular organization could be vastly different among foregut-derived tissues (larynx Vs esophagus Vs lung).

*Reviewer #2 (Recommendations for the authors):*

The text of the paper discussing the *Sox2* and Nkx2.1 data would be more straightforward if not presented in the context of asking whether dorsal cells spread ventrally in Shh mutants – just present it as looking at patterning. As stated, this framing just comes across as an unanswered question, with an interpretation of the data that isn't connected to the original question posed. Alternatively, the authors should do a better job of pointing out that this result didn't answer the question, and then use this to segue more smoothly into the next section on the Pax1-expressing cells.

The question of whether these Pax1-expressing abnormal epithelium cells come from the pouches is currently just observed, several possibilities are proposed, then it is left as an unknown. Given that there is already published data suggesting that thymus-identity cells spread into the pharynx in Shh mutants (Moore-Scott, et al), using regionalized pouch markers could be a nice and relatively easy way to address that question. Also a tight temporal series between E9.5-10.5 during pouch formation looking at Pax1 expression itself could be informative, Coronal sections and 3D imaging or reconstructions during these ages could also show this process more clearly.

*Reviewer #3 (Recommendations for the authors):*

1. Severe defects in Shh-/- mutants range from cadherin switching/distribution, the disintegration of basement membrane followed by extrusion of cells from the epithelial domain, and defects in cell survival. However, in wild-type vocal epithelium, that is undergoing remodeling, the defects are mild (increased RAB11 activity along with cadherin switching like events) in domains where Shh expression is low. The authors need to explain why low SHH expressing cells do not develop acute defects.

2. The authors observe that Foxa2 expression is reduced in epithelial cells in the absence of SHH signaling and suggest that Shh might prevent EMT by positively regulating Foxa2 autonomously or non-autonomously. The authors should clarify where Foxa2 is expressed in both control and mutant embryos, and attempt to resolve where FOXA2 may be acting.

3. The observation that epithelial cells are being replaced by a new population of cells is very interesting, but this part of the manuscript would benefit from additional clarification. The authors never do lineage tracing experiments to show that the cells are of pharyngeal origin, as Pax1 expression might suggest. Further documentation of how Pax1 expression is changed in control and mutant tissues, as well as the possible use of additional pharyngeal markers, might help clarify what is going on.

---

## [Author Response]

Reviewer #1 (Recommendations for the authors):That being said, a few issues need to be addressed.1. The authors emphasize EMT in foregut development. However, the evidence for EMT needs further supports. In Figure 2E, F why the mesenchyme is negative for N-cad, while the mesenchyme in 3P,Q is positive? In this reviewer's opinion, it is possible that deletion of Shh reduces the levels of matrix proteins that attach the epithelium to the basement, and the mutant epithelial cells become loosely attached. During the dynamic morphogenesis process these mutant cells are pushed away, not necessarily going through EMT. Once detached from the BM, these epithelial cells cannot survive and go through apoptosis. This could be the reason for so many cell deaths in the mutants.

We examined the expression of Vimentin and SNAIL which are required and sufficient for EMT induction and also mark the mesenchyme in controls. We focused on *Shh*-descendant cells at E9.75, prior to the onset of widespread cell death in order to get a more complete view of transitioning epithelial cells. We found that while SNAIL is not expressed within control (*Shh^CreER/+;^Rosa^Ai9/+^*) or mutant (*Shh^CreER/-;^Rosa^Ai9/+^*) epithelia, there were low levels of Vimentin expression along the apical surface of the mutant epithelium (approximately 23 TdT+ cells expressing Vimentin across 4 replicates) (Figure 3-supplement figure 3A-B,E). Additionally we found that a subset (~30-50%) of TdT+ cells found in the mesenchyme within mutants expressed SNAIL or Vimentin, suggesting that some of these cells express mesenchymal markers in addition to undergoing a cadherin-switch and de-laminating from the epithelial layer (Figure 3—figure supplement 3A-D, F). We were unable to examine additional mesenchymal markers in these cells (i.e. FOXF1 and *SOX9*) because their expression is dramatically reduced in *Shh^-/-^*s (Nasr *et al.*, 2019, 2021). Additionally, the sharp increase in cell death by E10.5 within the mesenchyme prevents us from capturing the entire population of TdT+ cells within the mesenchyme at this stage. Due to these limitations, we agree that it is difficult to rule out the possibility of an EMT-independent laminin-destabilization contributing to this phenotype. We have discussed this and our new EMT data extensively in the Discussion section entitled “HH-dependent epithelial changes are mediated by a partial EMT”.

Regarding the question about N-Cadherin expression in the mesenchyme, N-Cadherin is in fact expressed in all the mesenchymal tissues (wild-type and *Shh^-/-^*). However, the ectopic N-Cadherin in the epithelia of E10.5 mutants is much brighter than any of the mesenchymal staining and would be fully saturated at levels required to show the fainter mesenchymal expression. In contrast, ectopic N-Cadherin expression levels are initially low in mutant epithelia (Figure 3P,Q at E9.75 as well as throughout E10 – see other images in Figure 3), allowing us to image non-saturating expression in the mesenchyme and epithelia.

2. The authors suggested that Pax1+ cells expand in place of ShH^+^ cell derivatives. Although the lineage tracing in Figure 4E-H seems interesting, it is known that E10.5 foregut of Shh null mutants is severely disrupted. Is there any way to confirm that the section is taken at the same level as the wildtype control?

To address this, we performed additional experiments with wholemount embryos that coupled fluorescent in situ hybridization (HCR) with immunofluorescence. We identified *Hoxb1* as a marker of the larynx epithelium at E10.5 in both controls and *Shh^-/-^*s and used sections from the wholemounts to establish morphological parameters to identify the larynx epithelium. At E10.5, the anterior larynx epithelium is fused with that of the 4^th^ pharyngeal pouches on each lateral side, generating a single, wedge-shaped lumen that is wide at the dorsal end and narrow at the ventral end. At more posterior regions the pouch epithelia are distinct, flanking both sides of the larynx, which has a small circular lumen that is continuous with the rest of the gut tube. It is not possible to morphologically distinguish the posterior boundary of the larynx from that of the trachea and esophagus. Using this criteria, there were multiple images that were anterior to the larynx in the previous version of the manuscript. We replaced the images in panels H-J in Figure 4 as well as panels A-B in Figure 4—figure supplement 6 with sections through the larynx. We also replaced panels within Figure 2C-D, 3J-N, 4H-L, 5C-E, supplement 3-1C, and supplement 4-5D,G,H.

While the data from the sections in the correct plane was largely the same, there were two changes to our conclusions. First, instead of being completely absent from the mutant epithelium at E10.5, E-Cadherin is downregulated compared to controls (new Figure 2D). Second, *Pax1* expression does not extend into the mutant larynx epithelium (Figures 5F-Q and Figure 5—figure supplement 2). Interestingly, the new data shows that the 4^th^ pharyngeal pouches are adjacent to the laryngeal epithelium and in some sections appear to have fused with it to create a single epithelial tube (Figure 4—figure supplement 6, Figure 5P-Q). We have changed our discussion and our model to reflect that the 4^th^ pharyngeal pouch is the likely source of the ectopic epithelial cells (Figure 6B).

From proximal (the larynx) to distal (stomach) shh expression patten is dynamically changed. In addition, the staining for cleaved caspase 3 seems not very specific.

We address the concept of dynamic Shh expression within the larynx in the discussion (lines 419-429) where we note that multiple enhancers govern its expression:

“Confirming previous studies, we find that *Shh* is dynamically expressed during larynx development (Sagai *et al.*, 2009; Lungova *et al.*, 2015, 2018). In addition to its downregulation from regions of the larynx epithelium that express N-Cadherin, the relative levels of *Shh* within *Shh*-expressing domains of the epithelium are highly dynamic at later stages, where there is an overall reduction in *Shh* within the dorsal half of the larynx epithelium, which is contiguous with the esophagus, compared to the ventral half (Figure 1B,H). Additionally, lower levels of *Shh* have been reported in the trachea compared to the esophageal epithelium at later stages (Nasr *et al.*, 2021). *Shh* expression in the larynx is regulated by three distinct enhancers that occupy largely non-overlapping regions of activity along the dorsal-ventral axis of the larynx (Sagai *et al.*, 2009, 2017; Tsukiji, Amano and Shiroishi, 2014). While it remains unclear how they are regulated, differential enhancer utilization is a plausible mechanism for regional regulation of *Shh* along the foregut.”

We agree that the cleaved-caspase 3 staining is not specific to the laryngeal region of the foregut. As we do not make any claims about the specificity of the cell death, we have not added a specific statement summarizing cell death in other areas of the *Shh^-/-^* foreguts. Instead, we now comment on the role that cell death in the adjacent pharyngeal arches likely plays in changing the morphology of the posterior pouches.

(lines 344-348).“This fusion of the two compartments is likely due to high levels of cell death within both the vocal fold mesenchyme as well as throughout the adjacent pharyngeal arch mesenchyme (arches 2 and 3) (Moore-Scott and Manley, 2005), which we speculate result in the collapse of the 4^th^ pharyngeal pouches towards the midline of the foregut (Figure 4—figure supplement 5H, Figure 6B).”

3. Similar to the issue raised in 2, is there any way to make sure that the mutant and wildtype tissue sections shown in Figure 4—figure supplement 4B are taken at the same level? The cellular organization could be vastly different among foregut-derived tissues (larynx Vs esophagus Vs lung).

We have added two criteria to address this concern. First, we use the morphological criteria defined in our response to comment #2 above to determine the level of the sections taken for analysis of the epithelial layer. These new images confirm that the tissues originally shown in this figure as well as the additional data that was used for the analysis are in the plane of the larynx. Second, we used *Hoxb1*, which we have identified as a marker for the larynx epithelium (Figure 5G). We have replaced the original panels with new data showing both *Hoxb1* and ß-Catenin (new Figure 4—figure supplement 6A-B).

Reviewer #2 (Recommendations for the authors):The text of the paper discussing the Sox2 and Nkx2.1 data would be more straightforward if not presented in the context of asking whether dorsal cells spread ventrally in Shh mutants – just present it as looking at patterning. As stated, this framing just comes across as an unanswered question, with an interpretation of the data that isn't connected to the original question posed. Alternatively, the authors should do a better job of pointing out that this result didn't answer the question, and then use this to segue more smoothly into the next section on the Pax1-expressing cells.

We agree, and have revised this section as suggested. The text now states:

“We then examined the expression of the patterning markers *SOX2* and NKX2.1 (Que *et al.*, 2007; Kim *et al.*, 2019; Nasr *et al.*, 2019; Kuwahara *et al.*, 2020). At E10.5 there is an absence of NKX2.1 and a significant reduction in nuclear *SOX2*, which is undetectable by E11.5 (Figure 5A-E, Figure 5—figure supplement 1A-D). We conclude that larynx epithelial cells are abnormally patterned in *Shh^-/-^* embryos.”

The question of whether these Pax1-expressing abnormal epithelium cells come from the pouches is currently just observed, several possibilities are proposed, then it is left as an unknown. Given that there is already published data suggesting that thymus-identity cells spread into the pharynx in Shh mutants (Moore-Scott, et al), using regionalized pouch markers could be a nice and relatively easy way to address that question. Also a tight temporal series between E9.5-10.5 during pouch formation looking at Pax1 expression itself could be informative, Coronal sections and 3D imaging or reconstructions during these ages could also show this process more clearly.

We undertook the suggested experiments. We used whole-mount HCR coupled with immunofluorescence to examine the relative distribution of *Pax1* (normally expressed in pharyngeal pouches 1-3; (Müller *et al.*, 1996; Moore-Scott and Manley, 2005)) and *Hoxb1* (expressed in the 4th pharyngeal pouches; (Moore-Scott and Manley, 2005)) expression in control and *Shh^-/-^* foreguts at E10.25 (30-32s; Figure 5F-Q) prior to the onset of widespread cell death, as well as at E10.75 (37-38s), and E11.5 (42-44s) (new Figure 5—figure supplement 2). We then generated 3-dimensional renderings to compare expression domains between controls and mutants. Contrary to our initial assessment of this phenotype, we found that while *Pax1* expression expands in the anterior pharyngeal pouches, it does not extend to the level of the larynx (see Reviewer 1, response #2; new Figure 5P-Q)

To determine whether thymic progenitors populate the larynx in the absence of HH signaling, we examined *Foxn1* and *Pax1* expression in control and *Shh^-/-^* tissues at E11.5 (when the thymus is first specified and *Foxn1* expression is first detected; (Wallin *et al.*, 1996)). The *Foxn1* expression domain does not extend beyond the boundary of *Pax1* expression, suggesting that thymic progenitors do not spread to the level of the larynx in Shh-mutants at these stages (Figure 5—figure supplement 3).

The new data also suggests that the 4^th^ pharyngeal pouches fuse with the anterior larynx and are likely the source of the new cells populating the *Shh^-/-^* larynx. We made extensive alterations to the results, figures and discussion to incorporate this data (detailed in Reviewer 1, response #2).

Reviewer #3 (Recommendations for the authors):1. Severe defects in Shh-/- mutants range from cadherin switching/distribution, the disintegration of basement membrane followed by extrusion of cells from the epithelial domain, and defects in cell survival. However, in wild-type vocal epithelium, that is undergoing remodeling, the defects are mild (increased RAB11 activity along with cadherin switching like events) in domains where Shh expression is low. The authors need to explain why low SHH expressing cells do not develop acute defects.

We speculate that the comparatively minor changes in low-*Shh* expressing cells in the wild-type larynx epithelium during larynx-esophageal separation are due to the persistence of FOXA2, which is downregulated and lost in *Shh^-/-^* foreguts. We have added data showing that FOXA2 expression is maintained both at the lateral edges of the constricting esophagus as well as within the separating epithelial lamina in anterior and more posterior larynx sections (Figure 4—figure supplement 3). As FOXA2 is required to maintain epithelial integrity via regulation of E-Cadherin (Zhang *et al.*, 2015; Bow *et al.*, 2020; Scheibner *et al.*, 2021), we propose that this provides an explanation for why we do not see down-regulation of E-Cadherin within these remodeling cells (Figure 1—figure supplement 1F,G). We have included new language in the results and discussion discussing this new data and clarifying our model regarding FOXA2 regulation (see below).

Results:

“As epithelial remodeling in wild-type embryos occurred in low-*Shh* expressing cells (Figure 1, Figure 1—figure supplement 1,2), we asked if these regions also had downregulated FOXA2. In contrast to the early foregut (Figure 4A, Figure 4—figure supplement 1), FOXA2 was not downregulated at the site of larynx-esophageal separation or along the lateral edges of the constricting esophagus (Figure 1, Figure 4—figure supplement 3A-D-F3,G,H,I). We conclude that FOXA2 regulation by HH signaling is specific to early stages of foregut and larynx development.”

Discussion:

“In contrast to the phenotypes observed in germline mutants, the changes in cell properties of low-*Shh* expressing cells in the wild-type remodeling larynx during larynx-esophageal separation were comparatively mild. These cells do not undergo apoptosis, perhaps because of the presence of low-level HH signaling. Most prominently, they do not downregulate E-Cadherin as they do in *Shh^-/-^* larynx epithelia (Figure 1,2, Figure 1—figure supplement 1). One explanation for the milder phenotype likely lies in their differential regulation of FOXA2, which is downregulated in *Shh^-/-^* foreguts but is maintained during larynx-esophageal separation (Figure 4—figure supplement 1,3). *Cdh1* (encoding E-Cadherin) has been reported to be transcriptionally regulated by FOXA2 in gastrulating endoderm as well as oral and breast cancer cells (Zhang *et al.*, 2015; Bow *et al.*, 2020; Scheibner *et al.*, 2021). Thus, the reduction of FOXA2 from the endoderm in *Shh^-/-^*s could initiate changes in cell adhesion resulting from reduced production of E-Cadherin.”

2. The authors observe that Foxa2 expression is reduced in epithelial cells in the absence of SHH signaling and suggest that Shh might prevent EMT by positively regulating Foxa2 autonomously or non-autonomously. The authors should clarify where Foxa2 is expressed in both control and mutant embryos, and attempt to resolve where FOXA2 may be acting.

We find that FOXA2 expression is specific to the laryngeal and foregut epithelium (i.e. the trachea and esophagus and more posterior gut structures) but is absent from the pharyngeal pouches by whole-mount staining (Figure 5—figure supplement 1E-F) in controls. By comparison in *Shh^-/-^*s, we find that FOXA2 expression is gradually lost from both the pharyngeal and laryngeal epithelium, and is also reduced more posteriorly in the trachea and esophagus (Figure 5—figure supplement 1F). This indicates that this process is not limited to the larynx and instead suggests that SHH regulates the maintenance of FOXA2 throughout the anterior foregut epithelium (Discussion lines 350-353 provided below). We also extensively discuss the possibility that FOXA2 is being directly or indirectly regulated by SHH in the Discussion section entitled “Does HH signaling directly regulate epithelial genes?”.

Discussion lines 350-353:

"FOXA2 expression is also absent from the more posterior trachea and esophagus compared to controls (Figure 5—figure supplement 1F). This suggests that FOXA2 regulation by *Shh* extends throughout the anterior foregut and that posterior tracheal/esophageal tissues might also be replaced in *Shh^-/-^*s.”

3. The observation that epithelial cells are being replaced by a new population of cells is very interesting, but this part of the manuscript would benefit from additional clarification. The authors never do lineage tracing experiments to show that the cells are of pharyngeal origin, as Pax1 expression might suggest. Further documentation of how Pax1 expression is changed in control and mutant tissues, as well as the possible use of additional pharyngeal markers, might help clarify what is going on.

We agree and have performed experiments with additional markers using wholemount imaging. These have provided a better resolution and context for interpreting the *Shh^-/-^* embryos. We find that this new population of cells is actually not *Pax1*-expressing (see responses to Reviewer 1 #2 and Reviewer 2 #1). As described in those responses, we propose that pharyngeal pouch 4 is the source of this population of cells and have added this to our model in Figure 6A as well as to extensively revised results and Discussion sections.

References

Bow, Y.-D. *et al.* (2020) ‘Silencing of FOXA2 decreases E-cadherin expression and is associated with lymph node metastasis in oral cancer’, *Oral Diseases*, 26(4), pp. 756–765. Available at: https://doi.org/10.1111/odi.13282.

Kim, E. *et al.* (2019) ‘Isl1 Regulation of Nkx2.1 in the Early Foregut Epithelium Is Required for Trachea-Esophageal Separation and Lung Lobation’, *Developmental Cell*, 51(6), pp. 675-683.e4. Available at: https://doi.org/10.1016/j.devcel.2019.11.002.

Kuipers, D. *et al.* (2014) ‘Epithelial repair is a two-stage process driven first by dying cells and then by their neighbours’, *Journal of Cell Science*, 127(6), pp. 1229–1241. Available at: https://doi.org/10.1242/jcs.138289.

Kuwahara, A. *et al.* (2020) ‘Delineating the early transcriptional specification of the mammalian trachea and esophagus’, *eLife*, 9, p. e55526. Available at: https://doi.org/10.7554/*eLife*.55526.

Lungova, V. *et al.* (2015) ‘Ontogeny of the mouse vocal fold epithelium’, *Developmental Biology*, 399(2), pp. 263–282. Available at: https://doi.org/10.1016/j.ydbio.2014.12.037.

Lungova, V. *et al.* (2018) ‘Β-Catenin signaling is essential for mammalian larynx recanalization and establishment of vocal fold progenitor cells’, *Development*, p. dev.157677. Available at: https://doi.org/10.1242/dev.157677.

Moore-Scott, B.A. and Manley, N.R. (2005) ‘Differential expression of Sonic hedgehog along the anterior–posterior axis regulates patterning of pharyngeal pouch endoderm and pharyngeal endoderm-derived organs’, *Developmental Biology*, 278(2), pp. 323–335. Available at: https://doi.org/10.1016/j.ydbio.2004.10.027.

Müller, T.S. *et al.* (1996) ‘Expression of AvianPax1andPax9Is Intrinsically Regulated in the Pharyngeal Endoderm, but Depends on Environmental Influences in the Paraxial Mesoderm’, *Developmental Biology*, 178(2), pp. 403–417. Available at: https://doi.org/10.1006/dbio.1996.0227.

Nasr, T. *et al.* (2019) ‘Endosome-Mediated Epithelial Remodeling Downstream of Hedgehog-Gli Is Required for Tracheoesophageal Separation’, *Developmental Cell*, 51(6), pp. 665-674.e6. Available at: https://doi.org/10.1016/j.devcel.2019.11.003.

Nasr, T. *et al.* (2021) ‘Disruption of a Hedgehog-Foxf1-Rspo2 signaling axis leads to tracheomalacia and a loss of *Sox9*+ tracheal chondrocytes’, *Disease Models and Mechanisms*, 14(2), p. dmm046573. Available at: https://doi.org/10.1242/dmm.046573.

Ohsawa, S., Vaughen, J. and Igaki, T. (2018) ‘Cell Extrusion: A Stress-Responsive Force for Good or Evil in Epithelial Homeostasis’, *Developmental Cell*, 44(3), pp. 284–296. Available at: https://doi.org/10.1016/j.devcel.2018.01.009.

Que, J. *et al.* (2007) ‘Multiple dose-dependent roles for *Sox2* in the patterning and differentiation of anterior foregut endoderm’, *Development (Cambridge, England)*, 134(13), pp. 2521–2531. Available at: https://doi.org/10.1242/dev.003855.

Rosenblatt, J., Raff, M.C. and Cramer, L.P. (2001) ‘An epithelial cell destined for apoptosis signals its neighbors to extrude it by an actin- and myosin-dependent mechanism’, *Current Biology*, 11(23), pp. 1847–1857. Available at: https://doi.org/10.1016/S0960-9822(01)00587-5.

Sagai, T. *et al.* (2009) ‘A cluster of three long-range enhancers directs regional *Shh* expression in the epithelial linings’, *Development*, 136(10), pp. 1665–1674. Available at: https://doi.org/10.1242/dev.032714.

Sagai, T. *et al.* (2017) ‘Evolution of Shh endoderm enhancers during morphological transition from ventral lungs to dorsal gas bladder’, *Nature Communications*, 8(1), p. 14300. Available at: https://doi.org/10.1038/ncomms14300.

Scheibner, K. *et al.* (2021) ‘Epithelial cell plasticity drives endoderm formation during gastrulation’, *Nature Cell Biology*, 23(7), pp. 692–703. Available at: https://doi.org/10.1038/s41556-021-00694-x.

Tsukiji, N., Amano, T. and Shiroishi, T. (2014) ‘A novel regulatory element for Shh expression in the lung and gut of mouse embryos’, *Mechanisms of Development*, 131, pp. 127–136. Available at: https://doi.org/10.1016/j.mod.2013.09.003.

Wallin, J. *et al.* (1996) ‘Pax1 is expressed during development of the thymus epithelium and is required for normal T-cell maturation’, *Development (Cambridge, England)*, 122(1), pp. 23–30.

Zhang, Z. *et al.* (2015) ‘FOXA2 attenuates the epithelial to mesenchymal transition by regulating the transcription of E-cadherin and *ZEB2* in human breast cancer’, *Cancer Letters*, 361(2), pp. 240–250. Available at: https://doi.org/10.1016/j.canlet.2015.03.008.